# Organic Thermoelectric Materials as the Waste Heat Remedy

**DOI:** 10.3390/molecules27031016

**Published:** 2022-02-02

**Authors:** Szymon Gogoc, Przemyslaw Data

**Affiliations:** Centre for Organic and Nanohybrid Electronics, Silesian University of Technology, Konarskiego 22B, 44-100 Gliwice, Poland; szymon.gogoc@polsl.pl

**Keywords:** organic thermoelectric materials, thermoelectric effect, conjugated polymers, small organic molecules, carbon nanotubes

## Abstract

The primary reason behind the search for novel organic materials for application in thermoelectric devices is the toxicity of inorganic substances and the difficulties associated with their processing for the production of thin, flexible layers. When Thomas Seebeck described a new phenomenon in Berlin in 1820, nobody could have predicted the future applications of the thermoelectric effect. Now, thermoelectric generators (TEGs) are used in watches, and thermoelectric coolers (TECs) are applied in cars, computers, and various laboratory equipment. Nevertheless, the future of thermoelectric materials lies in organic compounds. This paper discusses the developments made in thermoelectric materials, including small molecules, polymers, molecular junctions, and their applications as TEGs and/or TECs.

## 1. Introduction

In recent years, there has been a significant increase in energy-saving construction and green energy production. Organizations such as the European Union are attempting to address the issues caused by global warming by investing in green energy production. Much effort is put into developing materials and systems that can be used to produce energy from renewable resources, but it is uncertain what happens to the lost energy. According to the IDTechEx report, approximately 60% of electricity is wasted as heat. However, the ways of avoiding heat loss or recovering heat are unknown. The new research goal is to recover and convert the lost energy into a more acceptable form, such as electricity. Nevertheless, this is possible only using an appropriate device, such as a thermoelectric generator (TEG) [1].

The history of thermoelectrics began on 14 December 1820, when Thomas Seebeck described a new phenomenon in a presentation at the Berlin Academy of Sciences based on the copper–bismuth junction. During heating, the magnetic needle in the compass pointed in a different direction, indicating the direction of an electric field, which confirmed the electric current flow. In short, Seebeck discovered the effect of converting a thermal gradient into electricity. The Seebeck effect can be explained by the below equation [2,3,4]:α = −dV/dT(1)

In a letter written to Seebeck in 1823, Hans Christian Oersted proposed the name “thermoelectrics” for a group of devices discovered by Seebeck [5].

In 1834, the French watchmaker Jean Peltier discovered a new thermoelectric effect contrary to the Seebeck effect. According to Peltier, electricity was converted into a thermal gradient, depending on the direction of the current flow. This effect allowed achieving both thermoelectric cooling and heating. The first Peltier device could lower the bismuth–antimony junction temperature by 4.5 K at an ambient temperature of 30 °C. The Peltier effect was confirmed four years later by Emil Lenz, a Russian scientist [6,7].

During the 20th century, several thermoelectric devices were developed for different applications. These include radio devices powered by kerosene lamps and space probes, such as for the Pioneer and Voyager programs, powered by a plutonium-238-fueled radioisotope TEG. The power produced by these generators was approximately 300 W, which was sufficient to send data about the solar system and beyond, due to the relatively high half-life of radioactive isotopes [8,9,10]. Meanwhile, Abram Fedorovich Ioffe proposed using thermoelectric coolers (TECs) instead of compressor refrigerators [11,12,13,14]. In the 1960s, Ioffe’s thermoelectric refrigerator had similar parameters to conventional ones. The junctions made for thermoelectric cooling had bismuth and tin in an 80:20 mass ratio. The Soviet-designed Peltier element consisted of two radiators: the first one had a surface area of 1.4 m^2^ and was used for cooling, while the surface area of the second one was 3.8 m^2^ to dissipate heat from the hot side of the device, which was an important breakthrough in those times. At an ambient temperature of 19 °C, the refrigerator modified by Ioffe could achieve a temperature drop of 21 K, consuming 40 W of power for 40 L of cooling chamber volume. In addition, this modified refrigerator was significantly more energy-efficient compared with the conventional refrigerator [11].

Thermoelectric devices had remained mostly unchanged for decades until organic molecules were found to exhibit similar behaviors [15]. According to the Scopus database, the first example of a small-molecule organic material (published in 1980) was methyl-ethyl-morpholinium-ditetracyanoquinodimethane (MEM(TCNQ)_2_), which had a temperature transition point of approximately 340 K. The conductivity parameter was strongly correlated with phase transition. Above that temperature, conductivity was 20 S∙cm^−1^, and conductivity decreased exponentially with a drop in temperature, down to 0.004 S∙cm^−1^. Moreover, after exceeding the transition point, conductivity was independent of the temperature. MEM(TCNQ)_2_ below the transition point achieved −65 μV∙K^−1^ of the Seebeck coefficient above the transition point, and it was constant regardless of temperature. Below the transition point, the Seebeck coefficient was dependent on temperature, and near 340 K, it was approximately zero. The lower the temperature was, the higher the Seebeck coefficient was. Nevertheless, the material’s potential power factor (σα2) was higher above the transition point, due to much higher conductivity. Higher conductivity resulted from the spin entropy of the charge carrier in the metallic system [16].

Small molecule materials were not only compounds considered for organic thermoelectrics. One of the first articles about polymeric thermoelectric devices was published in 1984 by Park et al. based on their observation of the thermoelectric phenomenon in polyacetylene derivatives [15]. Park et al. decided to dope polyacetylene with iron (III) chloride, compared with arsenic (V) fluoride and iodine, which were examined four years earlier, in 1980 [15,17]. Similar to the compound mentioned in the previous paragraph, it was possible to observe semiconductor–metal state transitions in polyacetylene, but this time they were obtained via the addition of dopant. Due to the high charge mobility in a metallic state, obtaining metallic state behavior was a goal for scientists. Doping was carried out by immersing polyacetylene film in nitromethane solution of FeCl_3_. Depending on the degree of doping, the Seebeck coefficient at room temperature was between 10.5 μV∙K^−1^ for the highest-doped polymer and 35.1 μV∙K^−1^ for the weakest-doped. Worth mentioning is the difference in conductivity. Despite having a 3.5 times lower Seebeck coefficient, *cis*-polyacetylene with the highest amount of dopant had nearly three orders of magnitude higher electrical conductivity (1.70 S∙cm^−1^ vs. 988 S∙cm^−1^), which had a crucial influence on produced power.

In comparison, polyacetylene doped with AsF_5_ resulted in a Seebeck coefficient of 9.0 μV∙K^−1^ when conductivity was 362 S∙cm^−1^, while iodine-doped polyacetylene achieved a Seebeck coefficient of 18.5 μV∙K^−1^ and 160 S∙cm^−1^ of conductivity. It was possible to conclude that polyacetylene behaves similarly to metallic samples. Moreover, polyacetylene films’ behavior matches quasi-one-dimensional transport, crucial for thermoelectric devices. Worth mentioning is that, despite the higher conductivity of *trans*-polyacetylene, it is *cis*-polyacetylene that is used for examining thermoelectrical parameters due to its ease of doping and, as a result, a higher field for improvement [15,17]. In 1989, conductivity was measured by the Montgomery method (used for measuring resistance changes in the transverse and longitudinal directions; electrodes in this technique are placed on the corners of a substrate [18]) for *cis*-polyacetylene doped with FeCl_3_, at approximately 30,000 S∙cm^−1^. One of the main challenges for organic thermoelectrics was the low conductivity of the first conducting polymers. The conductivity of polyacetylene can be varied, depending on the application, even by 12 orders of magnitude [17,19]. These two articles made it possible to observe cooperation amongst the pioneers of conducting polymers, such as Alan MacDiarmid, Alan Heeger, and Hideki Shirakawa [15,17]. This discovery caused a shift in the design of TEGs from heavy brick elements to thin-film elastic components, which is the most important advantage of organic thermoelectric devices (OTEs). Compared to inorganic devices, OTEs can maintain optimal parameters at much lower temperatures [15,19]. On the other hand, these are more sensitive to higher temperatures, as organic compounds decompose at such temperatures, while inorganic thermoelectric compounds remain stable up to 1000 K [20,21]. Therefore, the potential application of OTEs is considerably broader in comparison to classical devices. Furthermore, organic compounds and carbon materials are characterized by better mechanical parameters in terms of stretching and flexibility than inorganic substances, which gives a broader perspective for future applications [22,23].

The main goal of this review is to draw attention to novel organic thermoelectric materials in different fields as candidates for application in efficient devices and to reduce generated heat waste.

## 2. Organic Small-Molecular Thermoelectric Materials

### 2.1. Electron-Conductive Materials

One example of thermoelectric materials is small molecules with a conjugated system, which can modify the nature of electric conduction. Huang et al. investigated two compounds, namely aromatic-dicyanovinyl-dipyrrolo[3,4-c]pyrrole-1,4-diylidene)-bis(thieno[3,2-b]thiophene) (A-DCV-DPPTT) and quinoid-dicyano-methylene-dipyrrolo[3,4-c]pyrrole-1,4-diylidene)bis(thieno-[3,2-b]thiophene) (Q-DCM-DPPTT) (Figure 1), because they differed mainly in the system of double bonds in the main chain [24]. The compounds had high Seebeck coefficient values of over –1000 μV∙K^−1^. A-DCV-DPPTT without the dopant achieved a Seebeck coefficient (α) value of –1215 μV∙K^−1^, while the α value of Q-DCM-DPPTT was –874 μV∙K^−1^. However, the key to the usability of compounds such as TEGs is their electrical conductivity. Pure compounds have weak conductivities and hence require the use of *n*-type dopants, such as 4-(2,3-dihydro-1,3-dimethyl-1*H*-benzimidazole-2-yl)-*N*,*N*-dimethylbenzenamine (N-DMBI) in this case (Figure 1). With the addition of 5 wt% of this dopant, the device exhibited the most optimal thermoelectric parameters. At room temperature, the conductivity of A-DCV-DPPTT was 3.1 S∙cm^−1,^ and that of Q-DCM-DPPTT was 0.11 S∙cm^−1^, which corresponded to power factors (PF) of 95 and 1.7 W∙m^−1^∙K^−2^, respectively. At 373 K, the thermoelectric parameters of the compounds were higher because of the higher mobilities of charge carriers, which were electrons in both cases. The conductivities of A-DCV-DPPTT and Q-DCM-DPPTT at 373 K were 4.9 and 0.24 S∙cm^−1^, respectively, corresponding to PFs of 217 and 4.2 W∙m^−1^∙K^−2^, respectively (Table 1). Vertical thermal conductivity was also observed for A-DCV-DPPTT, which was determined to be 0.25 W∙m^−1^∙K^−1^ at room temperature and 0.34 W∙m^−1^∙K^−1^ at 373 K. This corresponded to 0.11 and 0.23 of the dimensionless figure of merit (ZT) for room temperature and 373 K, respectively. On the other hand, Q-DCM-DPPTT could not exhibit thermal conductivity due to its limited solubility in conventional solvents, which was a limiting factor for producing films with a thickness of more than 100 nm, as well as for the 3*ω*-technique used for measuring planar thermal conductivity. It is worth mentioning that the two investigated compounds differed in their doping mechanism. Q-DCM-DPPTT/N-DMBI was based on the classical system of doping, where free electrons are transferred from radicals to the host material. However, due to the low carrier mobility, which can be explained by the missing *π–π* stacking and the increased *d*-spacing distance, Q-DCM-DPPTT exhibited lower electrical conductivity, and thus weaker thermoelectric parameters, compared to A-DCV-DPPTT. On the other hand, due to its more regular structure and smaller *d*-spacing distance, the A-DCV-DPPTT/N-DMBI system positively influenced the charge transfer mobility [24].

Thermoelectric efficiency is determined not only by the correlation between dopant concentration and morphology, but the dopant type is also a critical factor in achieving appropriate thermoelectric parameters. Depending on the desired parameters, a *p*-type dopant can be used, which helps to obtain a positive Seebeck coefficient by increasing the number of hole carriers. By contrast, an *n*-type dopant improves the negative Seebeck coefficient applied by electrons [25,26]. Yuan et al. compared dopant types by using the 2-octyldodecyl derivative of two-dimensional π-expanded quinoidal terthiophenes (2DQTT-o-OD) (Figure 1) as a small-molecule compound for thermoelectric devices. The authors used 1,3-dimethyl-2,3-dihydro-1*H*-benzimidazoles with cyclohexyl groups (2-Cyc-DMBI-H, (2-Cyc-DMBI)_2_ and (2-Cyc-DMBI-Me)_2_) as dopants (Figure 1) [27]. They noted that pure 2DQTT-*o-*OD achieved a Seebeck coefficient value of –1000 μV∙K^−1^. The most optimal PFs were observed when added dopants were 10–15 mol%. The 2DQTT-*o-*OD with (2-Cyc-DMBI-Me)_2_ dopant exhibited the highest PF, which was measured as 17.2 μW∙m^−1^∙K^−2^ at room temperature, and 33.3 μW∙m^−1^∙K^−2^ at 363 K. Thermal conductivity measurements were carried out using the differential 3*ω*-technique, which showed a value of 0.28 W∙m^−1^∙K^−1^ for 2DQTT-*o-*OD doped by (2-Cyc-DMBI-Me)_2_, and subsequently, ZT was determined to be 0.02. Thermal conductivity could not be measured for the other tested samples due to the lack of anisotropic property. The authors also found that dimer dopants performed better because of their higher doping efficiency and produced twice the amount of anion radicals than 2-Cyc-DMBI-H [27].

**Table 1 molecules-27-01016-t001:** Thermoelectric parameters of *n*-type small-molecule materials.

Material	Dopant	Cond. Type	*HOMO* [eV]	*LUMO* [eV]	*σ* [S/cm]	*α* [μV/K]	*σα^2^* [μW/(mK^2^)]	ZT	Ref.
A-DCV-DPPTT	N-DMBI	N	−5.7	−3.9	3.1 ^(a)^ 4.9 ^(b)^	−568 ^(a)^−665 ^(b)^	95 ^(a)^ 217 ^(b)^	0.11 ^(a)^ 0.23 ^(b)^	[24]
Q-DCM-DPPTT	N-DMBI	N	−6.0	−4.5	0.11 ^(a)^ 0.24 ^(b)^	−383 ^(a)^ −420 ^(b)^	1.7 ^(a)^ 4.2 ^(b)^		[24]
2DQTT-*o*-OD	2-Cyc-DMBI-H	N		−4.68	0.18	387	2.7		[27]
2DQTT-*o*-OD	(2-Cyc-DMBI-Me)_2_	N		−4.68	1.1 ^(c)^	~550 ^(c)^	17.2 ^(a)^ 33.3 ^(c)^	0.02 ^(c)^	[27]
2DQTT-*o*-OD	(2-Cyc-DMBI)_2_	N		−4.68	0.43	409	7.2		[27]
TDPPQ	Bi (2 nm)	N			3.3 ^(d)^	−585 ^(d)^	113 ^(d)^		[28]
PDA		N	−6.47	−3.92		~600			[29]
PDI		N	−6.11	−3.58		~650			[29]
C_8_PDI		N				~4000			[29]
DNTT			−5.19	−1.82		~3.6 × 10^4^			[29]
C_10_DNTT						~1.2 × 10^5^	~0.5		[29]
C_8_BTBT						~7 × 10^4^			[29]
Fullerene C_60_		N				~−2 × 10^5^	~0.02		[29]
PTEG-1	N-DMBI	N			2.05	−248	16.7		[30]
PCBMNDI-CN	N-DPBI	N	−6.00 −6.66	−4.00 −4.60	~0.03	−1400	5		[31,32]
PDI-1, *n* = 2		N		−4.00	~1 × 10^−3^	~−200			[33]
PDI-2, *n* = 4		N		−4.00	~0.01	~−200			[33]
PDI-3, *n* = 6		N		−4.00	~0.5	~−200	1.4		[33]
NDI3HU-DTYM2		N			0.4	−250	2.5		[34]
NDI(2OD)(4tBuPh)-DTYM2		N			0.1	−187	0.35		[34]
Au-BP-Au		N				−6.9			[35]
Au-BDNC-Au		N				−13.3			[36]
Au-BDCN-Au		N				−11.5			[36]
Au-Sc_3_N@C_80_-Au						±20			[37]

^(a)^ Room temperature; ^(b)^ 373 K; ^(c)^ 363 K; ^(d)^ 333 K.

Not only organic compounds can be used for doping in OTEs. A common doping technique involves the addition of metals such as bismuth, which are characterized by adequate parameters to serve as *n*-type dopants. Huang et al. used bismuth as a dopant for thiophene-diketopyrrolopyrrole-based quinoidal (TDPPQ) (Figure 1), a compound that possesses potential thermoelectric properties. Pure TDPPQ had a Seebeck coefficient value of approximately −600 μV∙K^−1^. Using the vacuum deposition technique, the authors deposited bismuth on the film [28], and the main goal of their research was to optimize the temperature and the thickness of the bismuth layer. During measurements, the authors observed a significant decrease in the Seebeck coefficient for a bismuth layer thickness of over 2.5 nm, which was not compensated with the conduction of thin film. The results showed that a bismuth layer thickness of 2 nm was optimal for the Seebeck coefficient and conduction. At 333 K, the parameters of the device were found to be the highest: the Seebeck coefficient for a 2 nm bismuth layer on TDPPQ film was approximately −585 μV∙K^−1^, which was almost equal to that of the pure compound; electrical conductivity obtained by the film was 3.3 S∙cm^−1^; and the PF was 113 μW∙m^−1^∙K^−2^ (Table 1). Unfortunately, the ultrathin film could not exhibit thermal conductivity, and it was impossible to obtain ZT. However, low thermal conductivity and a high ZT can be expected for this thin film, compared with two-dimensional materials and materials with a low doping level. Due to the Fermi level shift, bismuth can be considered an effective interfacial dopant. As a result of that, it was possible to observe the change of *HOMO* level from 2.58 (pristine TDPPQ) to 2.73 eV below the Fermi level (TDPPQ with a 3 nm bismuth layer). Moreover, a possible gap state was approximately 0.60 eV below the Fermi level, allowing for an easier charge move between *HOMO* and *LUMO* levels [28].

PCBM, like any other material, also has some disadvantages. A critical drawback of PCBM is the aggregation of molecules in organic films, which negatively influences the conductivity of thin films. In the absence of aggregation, PCBM does not exhibit outstanding conductivity and requires the use of a dopant, namely 4-(1,3-dimethyl-2,3-dihydro-1*H*-benzimidazol-2-yl)-*N*,*N*-diphenylaniline (N-DPBI) (Figure 1), which is a derivative of N-DMBI having phenyl groups instead of methyl ones. Zuo et al. investigated whether the parameters of PCBM can be improved by adding core-cyanated naphthalene diimide (NDI-CN) (Figure 2). PCBM:NDI-CN could be used as an active layer for OPVs [31,32,38]. The authors spin-coated films at 600 rpm for 1 min and then at 3000 rpm for 20 s. With the addition of 5 wt% of dopant, an improved Seebeck effect was observed in the PCBM:NDI-CN blend (the Seebeck coefficient for 1 wt% addition of NDI-CN was up to −1400 μV∙K^−1^). As a result, the film’s conductivity was approximately 0.03 S∙cm^−1^, and the PF of the device was approximately 5 μW∙m^−1^∙K^−2^. However, the blend showed a decrease in the conductivity, which was more significant than the increase in the Seebeck coefficient. As a result, the PF also decreased, indicating the negative impact of the addition of NDI-CN to the PCBM film. In the studied system, due to the aggregation, NDI-CN acted as an agent that modifies the density of state only, not as a dopant, and negatively influenced conductivity [31,32].

Despite an unsuccessful attempt to utilize a naphthalene diimide derivative as a TEG material, Russ et al. continued their investigation on PDI derivatives. No dopant was used, due to the positive charge of nitrogen in the structure and hydroxyl anions, which act as counterions of nitrogen. These ions were created during the synthesis process and could transport charge efficiently. Three derivatives of PDI were synthesized (Figure 3) with different lengths of the side chain between the main structure and nitrogen, where the Seebeck coefficient was estimated at approximately −200 μV∙K^−1^ [33]. The most significant variations were noted in electrical conductivity. The conductivity of PDI-1 was slightly over 0.001 S∙cm^−1^, while that of PDI-2 was 0.01 S∙cm^−1^, which was one order of magnitude higher. PDI-3 had the highest conductivity of 0.5 S∙cm^−1^, and with a constant Seebeck coefficient, the highest PF of 1.4 μW∙m^−1^∙K^−2^. Unfortunately, the authors did not measure thermal conductivity and were thus unable to evaluate ZT. Their results suggested that increasing methylene spacer groups can positively influence the charge carrier mobility, resulting in improved conductivity [33].

### 2.2. Hole-Conductive Materials

Iodine-doped pentacene films exhibited excellent thermoelectric performance, with a Seebeck coefficient value of 70 μV∙K^−1^ and an electrical conductivity of 50 S∙cm^−1^. However, these films were characterized by poor stability in the air, and after 2 h, their conductivity was reduced by one order of magnitude. Fortunately, the decrease in conductivity could be lowered by repeated doping. The decrease in conductivity was attributed to the desorption of polyiodine from the film surface due to the formation of negatively charged iodine. Pentacene can be used as a candidate for observing the giant Seebeck effect (GSE), which is a phenomenon displayed by materials with a Seebeck coefficient value of at least 100 mV∙K^−1^ [29,39]. Nakamura et al. reported that GSE is one of the principles for developing thermoelectric devices. Most GSE materials are carbon nanotubes (CNTs) or their derivatives. However, semiconducting materials can also be used. Besides pentacene, GSE was observed in the following materials: sumanene, tetrabenzoporphyrin (BP) derivatives (including 5,15-didodecyltetrabenzoporphyrin, also known as C_12_BP), perylene diimide (PDI) derivatives, dinaphtho[2,3-*b*:2′,3′-*f*]thieno[3,2-*b*]thiophene (DNTT), which can be used for flexible devices, dioctylbenzothieno[2,3-b]benzothiophene, known as C_8_BTBT, and fullerene C_60_ (Figure 2) (Kojima et al., 2018). The main goal of research (Kojima et al., 2018) was to search for materials that can achieve a Seebeck coefficient value of at least 100 mV∙K^−1^ (Table 1 and Table 2) and exhibit electrical conductivity independently of *α*. The achievement of this goal allowed creating the prototypes of thermoelectric-generating fabric based on CNT derivatives [29,40,41,42]. In the study by Nakamura et al., the investigated compounds were deposited on glass substrates by applying the vacuum deposition technique under ultrahigh vacuum conditions (below 10^−7^ Pa). The highest Seebeck coefficient of about 200 mV∙K^−1^ was observed for fullerene, which fulfills the requirement of GSE. Two other materials that met the criterion for GSE were C_10_DNTT and BP, with Seebeck coefficient values of 120 and 100 mV∙K^−1^, respectively. The highest electrical conductivity was observed for BP, with a value of over 10^−6^ S∙cm^−1^, and thus, this material was characterized by the highest PF compared to other investigated materials. Nakamura et al. assumed that other materials had a thermal conductivity of 0.1 W∙m^−1^∙K^−1^ to calculate ZT directly from the PF, which for BP was approximately 0.005. The difference between the standard Seebeck effect and GSE is that they are based on different parameters (conductivity in the case of the standard Seebeck effect and the Seebeck coefficient in the case of GSE). Thus, the researchers focused on materials with much lower conductivity to make up the difference with a high Seebeck coefficient. The promising GSE materials should be characterized by high current mobility and low carrier concentration [29].

In addition to pentacene and fullerene, derivatives of carbon materials were used as the TEGs’ junction materials. Modified fullerene C_60_ with a triethylene glycol chain (PTEG-1) (Figure 2) was used as a host for an N-DMBI dopant (Figure 1) [30]. As in the case of bulk heterojunction (BHJ) organic photovoltaics (OPVs), in the apparent follow-up research, C_60_ was replaced by phenyl-C_61_-butyric acid methyl ester (PCBM). The results revealed an improvement in the Seebeck coefficient, which was −326 μV∙K^−1^ for PTEG-1 vs. −248 μV∙K^−1^ for PCBM, and electrical conductivity, which was 0.993 S∙cm^−1^ for PTEG-1 vs. 0.012 S∙cm^−1^ for PCBM. Both measurements were performed with the addition of 30 mol% dopants. However, only 40 mol% addition resulted in the optimal thermoelectric parameters of PTEG-1: −284 μV∙K^−1^ of thermoelectric power, 2.05 S∙cm^−1^ of electrical conductivity, and a PF of approximately 16.7 μW∙m^−1^∙K^−2^. ZT could not be calculated as thermal conductivity was not measured, but it can be assumed that PTEG-1 has similar conductivity to any other fullerene derivative in the undoped state, i.e., approximately 0.005 W∙m^−1^∙K^−1^. However, the change of conductivity with different dopant concentrations could be observed. It must be mentioned that the dopant molecules were inside the interlayer space and, therefore, did not affect the *π–π* stacking of PTEG-1, which led to an improvement in its electrical conductivity [30].

The results observed with PDIs encouraged scientists to conduct further studies on diimides. Zhang et al. investigated two naphthalene diimide derivatives, with two dithiolane rings attached to the core. The two compounds differed mainly in their side chains: NDI3HU-DTYM2 had two 3-hexylooctyl groups, while NDI(2OD)(4tBuPh)-DTYM2 had a 2-octyldecyl group and a 4-*tert*-butylphenyl group (Figure 3). The tested materials are used in organic field-effect transistor systems. For comparison, Zhang et al. doped poly(3-hexylthiophene) (P3HT) with graphene oxide and poly[2,5-bis(3-dodecylthiophen-2-yl)thieno[3,2*-b*]thiophene] (known as PBTTT-C_12_) with nitrosonium hexafluorophosphate [34]. The authors found that the studied diimides showed better thermoelectric parameters than P3HT and PBTTT-C_12_. NDI3HU-DTYM2 and NDI(2OD)(4tBuPh)-DTYM2 exhibited the maximum Seeback effect (Seeback coefficient: –600 and –400 μV∙K^−1^, respectively, vs. 390 μV∙K^−1^ for P3HT and 450 μV∙K^−1^ for PBTTT-C_12_). Similar to the Seebeck coefficient, the polymers also showed higher conductivities. For NDI3HU-DTYM2, the most optimal parameters were achieved with a conductivity of 0.4 S∙cm^−1^. The Seebeck coefficient was −250 μV∙K^−1^, and PF was approximately 2.5 μW∙m^−1^∙K^−2^, which is promising for use in the future. In the case of NDI(2OD)(4tBuPh)-DTYM2, the most optimal parameters were achieved with a conductivity of 0.1 S∙cm^−1^. The Seebeck coefficient was –187 μV∙K^−1^, and PF was calculated at approximately 0.35 μW∙m^−1^∙K^−2^ (Table 1). The authors could not calculate thermal conductivity due to the gate-modulated channel’s ultrathin nature; therefore, the characterization was based on the PF parameter instead of ZT [34].

Most cited studies have focused on TEGs, but some have also analyzed TECs. In both TEGs and TECs, the molecular junctions between the golden plate and golden probe (similar to those used in scanning tunnel microscopy) have been observed. Some examples of such systems are Au–BP–Au, Au–BPDT–Au, and Au–TPDT–Au, where BP refers to 4,4′-bipyridine, BPDT refers to biphenyl-4,4′-dithiol, and TPDT refers to terphenyl-4,4′-dithiol (Figure 4). These compounds are quite popular in organic synthesis. For example, bipyridine has been used as a ligand for reactions between organic substances and boron addition [35,43,44]. The Au–BP–Au, Au–BPDT–Au, and Au–TPDT–Au junctions presented an average Seebeck coefficient value of –6.9, 13, and 15.7 μV∙K^−1^, respectively (Table 1 and Table 2). These coefficients were better than the Au–Au junction alone, with a Seeback coefficient value one or two orders of magnitude lower at 0.25 μV∙K^−1^. Although molecular junctions showed weak performance as TEGs, the main goal of the research was to test their usability as TECs. For Au–BP–Au and Au–TBDT–Au, thermoelectric cooling could not be observed. However, Au–BPDT–Au showed a noticeable Peltier heat value of approximately 300 pW with 3 mV of applied potential. The measured value was not outstanding, but was sufficient to promote Peltier devices’ further development [35].

Other examples of molecular junctions that have been investigated are Au–BDA–Au, Au–BDNC–Au, Au–BDT–Au, and Au–BDCN–Au, where BDA refers to benzene-1,4-diamine, which is used as a dye; BDNC refers to benzene-1,4-diisocyanate, BDT refers to benzene-1,4-dithiol, and BDCN refers to benzene-1,4-dicyanate (Figure 4) [36]. The Seebeck coefficient values calculated for Au–BDA–Au, Au–BDNC–Au, Au–BDT–Au, and Au–BDCN–Au junctions were 2.2, −13.3, 2.4, and −11.5 μV∙K^−1^, respectively (Table 1 and Table 2). It is worth noticing that for every 0.35 μW consumed, 0.2 μW of heat was dissipated by Au–BDA–Au, 0.2 μW of heat by Au–BDNC–Au, 0.2 μW (calculated) of heat by Au–BDT–Au, and 0.25 μW (calculated) of heat by Au–BDCN–Au. These results support the development of thermoelectric devices [36].

The last example of molecular junctions is the cluster endohexal fullerene C_80_ derivative with scandium nitrite. In this junction, which was described as Sc_3_N@C_80_ (Figure 4), the scandium atoms interacted with carbon in a fullerene structure [37]. The thermoelectric parameters of this fullerene derivative were dependent on the distance between the needle and fullerene structure. The change of distance affected the thermoelectric performance of the junction, which in three cycles oscillated between 0 and 20, −5 and 10, and −20 and 0 μV∙K^−1^ (each oscillation applies to different molecules of Sc_3_N@C_80_). The observed results could be attributed to the tunneling effect due to the similarity in the conductivity of this molecular junction in different cycles [37].

## 3. Organic Polymer Thermoelectric Materials

### 3.1. Electron-Conductive Materials

Conducting polymers have been the subject of research worldwide due to their applications in organic electronics. One example of a conducting polymer is polydiimide derivative, which can be used in electrochromic devices and OPVs [45]. However, these compounds can also be used as thermoelectric materials. Wang et al. analyzed poly{[*N*,*N*′-bis(2-octyldodecyl)-naphthalene-1,4,5,8-bis(dicarboximide)-2,6-diyl]-*alt*-5,5′-(2,2′-bithiophene)}, known as P(NDI2OD-T2) (Figure 5), to examine the feasibility of using this polymer in TEGs. This compound is an excellent example of a donor–acceptor structure, in which the thiophene chain constitutes the donor and the diimide unit acts as the acceptor [46]. The Seebeck coefficient of pure P(NDI2OD-T2) was −795 μV∙K^−1^. To obtain more optimal parameters, the authors doped polydiimide by tetrakis(dimethylamine)ethylene (TDAE) (Figure 1), which caused an increase in electrical conductivity by one order of magnitude to 0.003 S∙cm^−1^ (Table 3). The Seebeck coefficient of doped polydiimide decreased to −208 μV∙K^−1^, resulting in a PF of 0.013 μW∙m^−1^∙K^−2^. Such a low PF may be related to contact geometry, which influences the measured conductivity [46].

The authors improved the parameters of P(NDI2OD-T2) by modifying its structure. Thiazole rings were added to the diimide, and the polymerization process was maintained with active zinc. This resulted in another donor–acceptor structure, known as poly{[*N*,*N*′-bis(2-octyldodecyl)-naphthalene-1,4,5,8-bis(dicarboximide)-2,6-diyl]-*alt*-5,5′-(2,2′-bithiazole)} P(NDI2OD-Tz2) (Figure 5). The resulting polydiimide was again doped with TDAE as was undertaken for P(NDI2OD-T2) to compare P(NDI2OD-T2) and P(NDI2OD-Tz2) [47]. The authors noted that P(NDI2OD-Tz2) exhibited higher conductivity than P(NDI2OD-T2) by one order of magnitude (0.06 S∙cm^−1^ vs. 0.003 S∙cm^−1^). The Seebeck coefficient of the pure material was found to be similar, but in the case of optimally doped material, the *α* value was twice higher (−447 μV∙K^−1^ vs. −208 μV∙K^−1^). The thermoelectric performance or PF of P(NDI2OD-Tz2) was two orders of magnitude higher than the unmodified polydiimide (0.013 μW∙m^−1^∙K^−2^ vs. 1.5 μW∙m^−1^∙K^−2^). No data are available regarding their thermal conductivity or ZT. Replacement of the thiophene ring with a thiazole ring allowed for better electron affinity, and thus, higher conductivity, which was sufficient to improve the TEG parameters of P(NDI2OD-T2) [47].

Another exciting example of a polydiimide derivative is poly{[*N*,*N*′-bis(2-triethyleneglicol)-naphthalene-1,4,5,8-bis(dicarboximide)-2,6-diyl]-*alt*-5,5′-(2,2′-bithiazole)} [P(gNDI-gT2)] (Figure 5). This compound has ethylene glycol units attached to thiophene rings and imide groups. Just like PNDI derivatives, this polymer had a donor–acceptor structure. The *n*-type dopant used for P(gNDI-gT2) was N-DMBI, a common dopant used for organic and polymeric structures. The main goal of the research on this compound was to achieve strong thermoelectric performance with the smallest possible addition of the dopant [49]. The earlier polymers showed dopant aggregation on AFM pictures. Pure P(gNDI-gT2) had a Seebeck coefficient value of −359 μV∙K^−1^; however, with the optimal addition of N-DMBI (10 mol%), a decrease of *α* to approximately −200 μV∙K^−1^ was noted. The electrical conductivity of the pure compound was approximately 0.1 S∙cm^−1^, which indicated a PF of 0.4 μW∙m^−1^∙K^−2^. With the optimal addition of dopant, the issue of aggregation was resolved, which was observed only with the addition of over 20 mol% dopant. Again, the material characterization was based on PF instead of ZT, due to the difficulty of measuring thin films’ thermal conductivity [49].

Wang et al. focused on the structure of polybenzimidazobenzophenanthroline (BBL) (Figure 6) because its main conjugation chain goes directly through the rings. This polymer did not have a donor–acceptor structure, and most importantly, showed phase transitions at high temperatures. The main application of BBL was as an active layer of OPVs, in which this polymer achieved 1.5% of photovoltaic conversion efficiency [46,57]. The Seebeck coefficient of pure BBL reached −400 μV∙K^−1^, which is a typical value for conducting polymers. Similar to P(NDI2OD-T2), BBL was doped with an *n*-type dopant, TDAE, which significantly increased its conductivity without a significant decrease in thermoelectric power. The Seebeck coefficient of BBL was inversely proportional to the fourth root of its conductivity. It is worth mentioning that P(NDI2OD-T2) and BBL were deposited on hexamethyldisilazane-modified glass. The maximum PF of BBL was approximately 0.43 μW∙m^−1^∙K^−2^, associated with a conductivity of 1 S∙cm^−1^ and a Seebeck coefficient of −60 μV∙K^−1^ [46].

Some attempts have been made to dope conducting polymers with fluoride anions. The main aim of such doping was to improve the conductivity of compounds by attaching fluoride ions into their main chain and transferring negative charges to the polymer. Chlorinated benzodifurandione-based poly(p-phenylene vinylene) (ClBDPPV) (Figure 6) was used with tetra-*n*-butylammonium fluoride (TBAF) (Figure 1) as a dopant. TBAF is a commonly used substrate for special applications, such as hydrogel synthesis with outstanding absorption parameters [52,58,59,60,61]. The thermoelectric power of pure ClBDPPV was approximately −1250 μV∙K^−1^. The addition of TBAF dopant up to 25 mol% caused an increase in the conductivity by five orders of magnitude, up to 0.62 S∙cm^−1^, but decreased the value of the Seebeck coefficient to −99.2 μV∙K^−1^. The PF obtained with this dopant concentration was approximately 0.63 μW∙m^−1^∙K^−2^. However, a noticeable drop in the conductivity value to 0.1 S∙cm^−1^ was observed after one week, which was caused by the affinity of *n*-type compounds to oxygen and the moisture in the air. Self-encapsulation of 1 μm thick film and the low *LUMO* level of ClBDPPV (approximately −4.28 eV) improved the stability of the device to a certain degree. Thermal conductivity was also evaluated for the doped film, which was estimated at 0.34 W∙m^−1^∙K^−1^. The authors who investigated ClBDPPV did not calculate ZT, but it can be assumed that measurements were made at room temperature, based on which, ZT can be calculated to be approximately 5.5 × 10^−4^ [52].

The thermoelectric performance is influenced not only by dopants but also by halogen groups in the thiophene chain. Yang et al. compared two polymers, poly[2,5-bis(2-octyldodecyl)-3,6-di(pyridin-2-yl)-pyrrolo[3,4-c]pyrrole-1,4(2H,5H)-dione-alt-(*E*)-2,2′-(ethene-1,2-diylbis(thiophene-5,2-diyl))] (PDPH), which has a donor–acceptor structure, and poly[2,5-bis(2-octyldodecyl)-3,6-di(pyridin-2-yl)-pyrrolo[3,4-c]pyrrole-1,4(2*H*,5*H*)-dione-alt-(*E*)-2,2′-(ethene-1,2-diylbis(3,4-difluorothiophene-5,2-diyl))] (PDPF) (Figure 6). These compounds underwent thermic degradation at approximately 370 °C and could be applied in TEGs. In both compounds, N-DMBI played the role of the donor [53]. The addition of N-DMBI to the polymer changed the characteristic of conduction. A dopant concentration of 9 wt% in PDPH resulted in a Seebeck coefficient value of −87 μV∙K^−1^, but with over 11 wt% dopant addition, the Seebeck coefficient was positive and reached 71 μV∙K^−1^, and the electrical conductivity with 13% dopant was 1.01 × 10^−3^ S∙cm^−1^. Under optimal conditions, PDPH achieved a PF of 5.11 × 10^−4^ μW∙m^−1^∙K^−2^. The introduction of fluorine into the polymer structure gave an impressive effect. PDPF with a 2 wt% addition of N-DMBI had a Seebeck coefficient value of –686 μV∙K^−1^, while the optimal parameters were achieved with 9% dopant addition. The optimized PDPF had a PF of 4.65 μW∙m^−1^∙K^−2^, which was four orders of magnitude higher than that of unmodified PDPH, the Seebeck coefficient was approximately −240 μV∙K^−1^, and the electrical conductivity was 1.3 S∙cm^−1^. PDPF exhibited better thermoelectric performance due to higher electron affinity, allowing more efficient doping. Less dopant was sufficient for obtaining more optimal parameters of this compound. Moreover, PDPF was more stable in the air than PDPH; the conductivity of fluorinated polymer was decreased by 26% in two weeks, while PDPH lost nearly 50% of its conductivity in the air during the same duration [53].

All of the previous results for polymers were mostly regarding TEGs that possessed the Seebeck effect. Theoretically, materials with this effect can be used as Peltier elements for thermoelectric cooling. Poly(1,1,2,2,-ethenetetrathiolates) (poly(A_x_[M-ett])) (Figure 7) is one of the polymers with a metal core that could be mostly copper or nickel. As the metal-core synthesis method was used, the polymer contained cations of metals such as potassium, sodium, nickel, or copper. The first thermoelectric polymers with metal-core (one of which was poly(nickel 1,1,2,2-ethenetetrethiolate) with potassium ion) were synthesized and investigated in 2012 [54,55,56]. The Seebeck coefficients of the tested compounds were average, ranging from −121.6 μV∙K^−1^ for poly(K_x_[Ni-ett]) to −75 μV∙K^−1^ for poly(Na_x_[Ni-ett]) and 83 μV∙K^−1^ for poly(Cu_x_[Cu-ett]) at room temperature. These polymers exhibited good electrical conductivity (44, 40, and 9.5 S∙cm^−1^ for poly(K_x_[Ni-ett]), poly(Na_x_[Ni-ett]), and poly(Cu_x_[Cu-ett]), respectively). The parameters resulted in relatively high PFs of compounds. By contrast, the highest PF was found to be approximately 66 μW∙m^−1^∙K^−2,^ which was determined for poly(K_x_[Ni-ett]) and comparable with the best polymers identified in the study. The thermoelectric cooling parameters were tested, but thermoelectric cooling was not observed for a device made from 35 thermocouples. However, the Peltier effect was observed in one based on poly(K_x_[Ni-ett]), and had temperature difference of 3.5 K at 0.6 V. The possible reason for this is the size of this device, which may allow faster heat dissipation and thus did not exhibit a visible Peltier effect. Interestingly, a similarly built TEG based on poly(Na_x_[Ni-ett]) produced 0.26 V of open voltage and 10.1 mA of short-circuit current, which indicates that with 200 thermocouples, the device can produce 1.5 V, which is a common voltage for most small electronic appliances [54,55,56].

Poly{[*N*,*N*′-bis(2-triethyleneglicol)-naphthalene-1,4,5,8-bis(dicarboximide)-2,6-diyl]-*alt*-5,5′-[2,2′-bis(2-dodecyloxythiophene)]} (PNDI2TEG-2T) (Figure 5) is a novel polydiimide with a donor–acceptor structure formed for applications in thermoelectric devices. Similar to P(NDI2OD-T2), the structure of PNDI2TEG-2T was modified to increase charge mobility. Liu et al. synthesized poly{[*N*,*N*′-bis(2-triethyleneglicol)-naphthalene-1,4,5,8-bis(dicarboximide)-2,6-diyl]-*alt*-5,5′-[2,2′-bis(octyloxythiazole)]} (PNDI2TEG-2Tz) (Figure 5). N-DMBI (Figure 2) was used as the dopant for both compounds. The difference between the two polydiimides was observable in atomic force microscope (AFM) pictures due to the more regular thickness of the polymer film [48]. The dopant added to PNDI2TEG-2T changed the carrier of electrical conduction from electron to hole, and the doped compound had a Seebeck coefficient value of −254.7 μV∙K^−1^ with 7 mol% addition of N-DMBI. However, with the increasing addition of dopant, the value of *α* changed from negative to positive, which was 66.8 μV∙K^−1^ for 56 mol% addition (Table 4). The highest PF of PNDI2TEG-2T was achieved with 42 mol% N-DMBI addition. The Seebeck coefficient value was positive and reached 57.2 μV∙K^−1^, while conductivity was estimated at 0.0007 S∙cm^−1^. Thus, PF was approximately 2.3 × 10^−4^ μW∙m^−1^∙K^−2^. Compared to the modified polymer, the parameters of PNDI2TEG-2T were found to be very poor. PNDI2TEG-2Tz maintained electron conduction regardless of the addition of dopant and exhibited strong thermoelectric performance with a Seebeck coefficient of approximately −263.1 μV∙K^−1^ for 7 mol% addition of dopant. The conductivity of this polymer was also two orders of magnitude higher with this amount of dopant addition. Optimal thermoelectric parameters were noted with 21 mol% addition of dopant. The Seebeck coefficient achieved with this amount of dopant was –159 μV∙K^−1^, and conductivity was 1.8 S∙cm^−1^, which was four orders of magnitude higher. The PF of PNDI2TEG-2Tz was approximately 4.6 μW∙m^−1^∙K^−2^, which was 20,000 times greater than for PNDI2TEG-2T (Table 3). The authors did not provide any data about thermal conductivity and ZT. Similar to the other pair of diimides investigated, PNDI2TEG-2T and PNDI2TEG-2Tz differed in their electron affinity, which allowed enhancing the conductivity and thus improving thermoelectrical parameters [48].

### 3.2. Hole-Conductive Materials

Most polydiimides are electron semiconductors, which have a negative Seebeck coefficient. However, these compounds were connected with *p*-type semiconductors, such as P3HT, resulting in block copolymers. Such copolymers can potentially provide electron and hole conductivity, depending on the doping type. One example of a block copolymer is poly(3-hexylthiophene)-*block*-{[*N*,*N*′-bis(2-triethyleneglicol)-naphthalene-1,4,5,8-bis(dicarboximide)-2,6-diyl]-*alt*-5,5′-[2,2′-bis(2-dodecyloxythiophene)]} [P3HT-*b*-P(NDI2OD-T2)], in which P(NDI2OD-T2) is attached with a compound that is popular in organic electronics such as OPVs and *p*-type TEGs (Figure 5). The copolymer was synthesized by Stille reaction, using P3HT with bromide end-groups, a diimide derivative with bromide groups, and a dithiophene unit with trimethyltin groups as substrates. N-DMBI was used as an *n*-type dopant, while 2,3,5,6-tetrafluoro-7,7,8,8-tetracyanoquinodimethane (F_4_TCNQ) (Figure 2) was used to improve hole conductivity [50]. The sign of the Seebeck coefficient changed depending on the dopant used. The maximum Seebeck coefficient values with 0.5 wt% dopant addition were −602 and 596 μV∙K^−1^ for N-DMBI and F_4_TCNQ, respectively. The electron-doped copolymer exhibited optimal thermoelectric parameters with 0.5 wt% dopant addition: conductivity of 1.7 × 10^−4^ S∙cm^−1^ and PF of 6.6 × 10^−3^ μW∙m^−1^∙K^−2^. The authors did not measure the exact thermal conductivity. However, it was observed that *p*-type doping required a higher concentration of dopant, and therefore, the optimal thermoelectric performance was obtained with 10 wt% addition of F_4_TCNQ. With this dopant concentration, the Seebeck coefficient value was reduced to 196 μV∙K^−1^, but conductivity increased to 1.4 × 10^−3^ S∙cm^−1^. The PF reached a value of 5.5 × 10^−3^ μW∙m^−1^∙K^−2^. Despite average results, it seemed that both *n*-type and *p*-type conductivity could be obtained from the same compound. The reason for the change in the characteristic of conductivity is the kind of block that forms the polymer. The addition of *p*-type dopant could increase the hole mobility of P3HT blocks, which enables changing the conduction type. The same applies to *n*-type dopants [50].

Dopants of a similar type were also compared to optimize the potential thermoelectric device. For instance, N-DMBI and N-DPBI were tested to assess their usability in polymer doping. N-DMBI has methyl groups in its structure, whereas N-DPBI (Figure 1) has phenyl ones. To compare different dopants, the polymer P(NDI2OD-T2) (Figure 5) was used [51]. Dopants were dissolved in the polymer solution to obtain a 9 mol% concentration. The thermoelectric performance of both N-DMBI- and N-DPBI-based devices was found to be similar. However, a slight difference in the Seebeck coefficient (−850 and −770 μV∙K^−1^ for N-DMBI- and N-DPBI-doped polymer) and conductivity (0.008 and 0.004 S∙cm^−1^ for N-DMBI- and N-DPBI doped polymer) indicated that the dopant with methyl groups was more advantageous. P(NDI2OD-T2) doped by N-DMBI had a PF three times higher (6 × 10^−7^ μW∙m^−1^∙K^−2^ vs. 2 × 10^−7^ μW∙m^−1^∙K^−2^ for N-DPBI-doped polymer) (Table 4). N-DMBI and N-DPBI exhibited poor solubility in the host (approximately 1% of dopant was in solution, while the rest aggregated on the surface), which may also have some advantages. However, such a low amount of soluble dopant will not significantly influence the sample morphology [51].

Carbazole materials can be used for many different applications in organic electronics, such as in organic light-emitting diodes (OLED) or organic capacitors. Poly[*N*-9′-heptadecanyl-2,7-carbazole-*alt*-5,5-(4′,7′-di-2-thienyl-2′,1′,3′-benzothiadiazole)] (PCDTBT) (Figure 8), a carbazole material, was tested to examine the possibility of using it as a thermoelectric material. This polymer was doped with iron (III) chloride as a *p*-type dopant [62,63]. The polymer exhibited high conductivity, which could allow outstanding thermoelectric performance. The electrical conductivity of doped PCDTBT reached up to 160 S∙cm^−1^ with a Seebeck coefficient value of 34 μV∙K^−1^ (Table 4). The authors who analyzed PCDTBT decided to evaluate its PF instead of its thermal conductivity due to difficulties determining this parameter. They found that the polycarbazole derivative had a PF of 19 μW∙m^−1^∙K^−2^, which confirmed its promising application as a thermoelectric material. Furthermore, the material seemed to be stable in air, allowing other potential applications of this compound. Moreover, PCDTBT did not require mechanical or thermal treatment, which may be considered as an added advantage [62,63].

In talking about conjugated polymers used in thermoelectric devices, it is hard not to mention poly(3-ethylenedioxythiphene) polystyrene sulfonate (PEDOT:PSS) (Figure 9). According to the Scopus database, PEDOT has been used as an object of research since 2008, when Jiang et al. conducted studies on PEDOT:PSS and its potential use in thermoelectric. The results obtained by this group gave perspective for improvements in the performance of PEDOT-based devices [64]. The Jiang group measured PEDOT:PSS (baytron P) with eight different treatments (additives) and preparation conditions and compared them with pristine PEDOT:PSS. When the dimethyl sulfoxide (DMSO—5% and 10%) and ethylene glycol (EG—5% and 10%) were used and measured in the range of 150 K up to 310 K, the Seebeck coefficient was between 12 and 16 μV∙K^−1^ at 310 K, and 8 to 12 μV∙K^−1^ at 150 K was obtained. The highest conductivity was measured for PEDOT:PSS with 5% EG annealed at 150 °C (42 S∙cm^−1^ in 150 K, 56 S∙cm^−1^ in 310 K). As for the figure of merit (ZT), the lowest value was for pristine PEDOT:PSS (0.2–0.4 × 10^−4^) and the highest with 5% of DMSO (without baking, 225–295 K with ZT = 0.0165) and 5% of EG (with baking, 150–225 K ZT = 0.018). The difference between pristine PEDOT:PSS and solvent treated materials would be caused by dedoping by DMSO and EG [64].

Optimization of preparation parameters is one of the key factors in device production. Huang et al. examined the influence of the annealing parameter on thermoelectric parameters. In the case of PEDOT:PSS, annealing can stack polymer in two possible orientations: edge-on and face-on. Both possess one-dimensional charge transport instead of three-dimensional, which improves the mobility of charge. The group compared Clevios PEDOT:PSS (PH1000) before and after annealing in the range of 200–250 °C. A 10 µm film was produced by vacuum filtration and vacuum drying. Then, the film was annealed for 12 h at the desired temperature. The carrier mobility improved due to annealing from 9.37 × 10^−3^ cm^2^∙V^−1^∙s^−1^ to 1.27 cm^2^∙V^−1^∙s^−1^ at 220 °C, which proved the most optimal temperature of annealing. Conductivity improved from 1–3 S∙cm^−1^ to 596 S∙cm^−1^ for the sample annealed at 220 °C at room temperature. The Seebeck coefficient also exceeded base parameters and reached 23.3 μV∙K^−1,^ and *PF* was 32.5 μW∙m^−1^∙K^−2^. AFM showed an increment of roughness and morphology changes, proving a crystallinity change between pristine and annealed PEDOT:PSS films [65].

The figure of merit is one of the most critical parameters describing thermoelectric materials. In 2013, Kim et al. adopted a different strategy than most researchers in TEG research. Kim decided to dedope the PEDOT:PSS and to optimize the thermoelectric parameters to obtain a ZT as high as possible. The hydrophilic nature of PSS and hydrophobic nature of PEDOT explains why researchers decided to use hydrophilic solvents—ethylene glycol (EG) and dimethyl sulfoxide (DMSO)—to improve conductivity and selectively dedope PSS. Instead of producing pellets, in 2008, the dedoped compound was spin-coated on a silicon substrate (4000 RPM, 30 s) and annealed on a hot plate at 130 °C for 15 min. The authors decided to observe the influence of the duration of the dedoping process on conductivity, the Seebeck coefficient, the power factor, thermal conductivity, and the figure of merit. The highest Seebeck coefficients were measured for PEDOT:PSS after 120 min of treatment (72 μV∙K^−1^ for DMSO) and after 300 min of exposure on EG (64 μV∙K^−1^). Regarding the conductivity, the maximum value for PEDOT:PSS with DMSO was achieved after 180 min of treatment and reached approximately 950 S∙cm^−1^. For EG-mixed samples, the maximum conductivity was up to 890 S∙cm^−1^ after 120 min of dedoping. The most optimal parameters for both treatments were for 120 min in DMSO, resulting in a Seebeck coefficient of 72 μV∙K^−1^, 880 S∙cm^−1^ of conductivity, a power factor of 469 μW∙m^−1^∙K^−2^, 0.24 W∙m^−1^∙K^−1^ of cross-plane thermal conductivity, 0.42 W∙m^−1^∙K^−1^ of in-plane thermal conductivity, and a figure of merit of 0.42. In EG treatment after 120 min, the values were a Seebeck coefficient of 62 μV∙K^−1^, 890 S∙cm^−1^ of conductivity, a power factor of 345 μW∙m^−1^∙K^−2^, 0.225 W∙m^−1^∙K^−1^ of cross-plane thermal conductivity, 0.52 W∙m^−1^∙K^−1^ of in-plane thermal conductivity, and a figure of merit of 0.28 [66].

Currently, PEDOT:PSS is still a subject of interest for flexible organics, which is shown by Xu et al. Pristine PEDOT has relatively poor thermoelectric parameters due to an excess of PSS, which increases the tunneling distance. Consequently, to utilize PEDOT:PSS as a thermoelectric material, this compound needs to be treated to decrease the excess PEDOT and to optimize the parameters. Therefore, Xu et al. decided to conduct a two-step post-treatment. The first step relies on polar solvents such as concentrated sulfuric acid and formamide (which were used in Xu et al.’s research), which can remove the excess PSS, and increase the conductivity without any significant changes in the Seebeck coefficient. The second step is to optimize conductivity and the Seebeck coefficient to obtain the maximum peak of the power factor. For this purpose, sodium borohydride was used. The main goal was to decrease the conductivity with an increment in the Seebeck coefficient to obtain an optimal power factor. In this way, PEDOT:PSS exhibited a Seebeck coefficient of 28.1 μV∙K^−1^ and 1786 S∙cm^−1^ of conductivity, and as a result, the power factor was approximately 141 μW∙m^−1^∙K^−2^. However, this compound’s thermal conductivity was not measured due to material tests in more practical conditions. Therefore, Xu et al. decided to prepare a flexible device based on treated PEDOT:PSS on polyimide substrates. That device produced 2.9 mV of voltage and approximately 1 μW∙cm^−2^ of power density (power produced per area unit). That means that PEDOT:PSS treated with sulfuric acid, formamide, and sodium borohydride can perform well as an active compound in flexible electronics [67].

Flexible devices are an obvious application for PEDOT:PSS. However, it is possible to obtain a different form of this compound for use in TEG. The Yang group prepared ternary composite fibers based on PEDOT:PSS, polyvinyl alcohol (PVA), and telluride nanowires (Te-NWs). Dedoping was necessary to improve the power output from the polymer; as a result, a small addition of ethylene glycol (7%_v_) was necessary. The mixture was dissolved in deionized water in the next step, and PVA was added. The main goal of using PVA was to improve the stretchability of the fibers compared with non-PVA fibers. Te-NWs were prepared from glucose and sodium tellurite (Na_2_TeO_3_) in the reactor for 12 h at 120 °C. The product was filtrated and dried in a vacuum at 50 °C. After 2 h of heating at 90 °C, a certain amount of prepared Te-NWs and 1 M sulfuric acid was put into the mixture to obtain a gel. The gel was put into a polytetrafluorene capillary for 3 h at 90 °C to obtain fiber, and then, those composites were bathed in EG to enhance their thermoelectric properties. The best thermoelectric performance resulted from the addition of 10% of PVA and 35% of Te-NWs to PEDOT:PSS after EG posttreatment, resulting in a Seebeck coefficient of 11.3 μV∙K^−1^, 382.4 S∙cm^−1^ of electrical conductivity, and a PF of 8.5 μW∙m^−1^∙K^−2^. The Yang group assembled a fiber-based device from ten pairs of composite and copper, which showed fair output voltage (5.03 mV) and power density (28.87 μW∙m^−2^) at a 60 K temperature difference. Those kinds of fibers have shown sufficient flexibility and thermoelectric parameters to be used in wearable thermoelectric generators [68]. The example discussed above is not the only way to connect organic compounds with inorganic nanoparticles. Bismuth telluride (Bi_2_Te_3_) is one of the most commonly used materials in thermoelectric generators due to its high figure of merit [80,81]. Wang’s group decided to prepare flexible devices based on the PEDOT/Bi_2_Te_3_ system. The bismuth telluride layer was produced by the nanosphere lithography method because of its low cost and high reproducibility. The key to a well-prepared layer of PEDOT/Bi_2_Te_3_ nanoparticles was adapting polystyrene nanospheres in order to obtain the desired diameter of nanoparticles. Chromium was used in the next step for producing a protective layer. Thermal deposition of Bi_2_Te_3_ and the dissolving of silicon oxide template were preparation steps for the vapor-phase polymerization of PEDOT. In this way, the production of nanospheres of three different diameters (100, 300, and 600 nm) was possible. During the tests of those materials, 100 nm PEDOT/Bi_2_Te_3_ (31%_v_) showed the best performance. Its Seebeck coefficient was 168 μV∙K^−1^ with 483 S∙cm^−1^ of electrical conductivity at 1350 μW∙m^−1^∙K^−2^ of power factor. Thermal conductivity measurements were also conducted and resulted in 0.71 W∙m^−1^∙K^−1^ of in-plane thermal conductivity. In this case, the calculation of the figure of merit was possible—the 100 nm PEDOT/Bi_2_Te_3_ (31%_v_) system possessed a ZT of 0.58. The main advantages of this material were the high power factor, the figure of merit, and mechanical flexibility, which make the preparation of high-efficient flexible thermoelectric generators based on these materials (organic-inorganic hybrids) possible. The reason for the materials’ performance was the interfacial effect caused by the optimal interfacial surface-to-volume ratio [69].

The doping method is also an important factor in modifying the electrical properties of organic compounds and conducting polymers. A dopant can be introduced by dissolving it in the compound solution and spin-coating it or via vapor deposition. An experiment was performed to optimize the thermoelectric parameters of poly(2,5-bis(3-alkylthiophen-2-yl)thieno[3,2-*b*]-thiophene) (PBTTT) derivative, which was also known as poly[2,5-bis(3-tetradecylthiophen-2-yl)thieno[3,2-*b*]thiophene] (PBTTT-C_14_) (Figure 7). Eventual optimization was influenced not only by the doping technique but also by the dopant type. Patel et al. compared two compounds, F_4_TCNQ and 2,5-difluoro-7,7,8,8-tetracyanoquinodimethane (F_2_TCNQ) (Figure 1). The main difference between these two dopants was the number of fluorine atoms in their structure [70]. The results revealed that the vapor-doping technique was better than spin-coating. The solution technique resulted in conductivity two orders of magnitude lower for F_4_TCNQ and five orders of magnitude lower in the case of F_2_TCNQ. This effect cannot be related to the simple difference in the Seebeck coefficient. The most optimized results were observed for F_2_TCNQ, with 36 S∙cm^−1^ of electrical conductivity, a Seebeck coefficient of 140 μV∙K^−1^, and a PF of 70 μW∙m^−1^∙K^−2^. The thermoelectric performance of vapor-doped PBTTT-C_14_ was the best in the study [70].

The study of the first copolymers of poly(2,5-dimethoxyphenylenevinylene-*co*-phenylenevinylene) [P(MeOPV-*co*-PV)] and poly(2,5-diethoxyphenylenevinylene-*co*-phenylenevinylene) [P(EtOPV-*co*-PV)] proved that not only polythiophene and polydiimide derivates could be used as thermoelectric materials (Figure 10). Poly(2,5-dimethoxyphenylenevinylene) and poly(2,5-diethoxyphenylenevinylene) copolymers with polyphenylenevinylene were *p*-type semiconductors with iodine as a dopant. Both polymers were stretchable enough (four times the basic dimensions) to be used in elastic thermoelectric devices [71,72]. The polymers were also characterized by excellent electrical conductivity (183 and 350 S∙cm^−1^ for P(MeOPV-*co*-PV) and P(EtOPV-*co*-PV), respectively). Although the copolymers had average Seebeck coefficients (43.5 and 47 μV∙K^−1^ for P(MeOPV-*co*-PV) and P(EtOPV-*co*-PV), respectively), they had strong PF values (34.6 μW∙m^−1^∙K^−2^ for iodine-doped P(MeOPV-*co*-PV) and 77.3 μW∙m^−1^∙K^−2^ for iodine-doped P(EtOPV-*co*-PV)) (Table 2). The thermal conductivity of P(MeOPV-*co*-PV) was determined to be 0.8 W∙m^−1^∙K^−1^, while that of P(EtOPV-*co*-PV) was 0.25 W∙m^−1^∙K^−1^. However, ZT was very high for P(EtOPV-*co*-PV) (0.1) and low in the case of P(MeOPV-*co*-PV) (0.014) at room temperature due to high thermal conductivity [71,72].

As a new kind of material exhibiting a high Seebeck effect, *p*-type polymer blends with a PF of 1000 μV∙K^−1^ were developed. Zuo et al. prepared two polymer blends based on P3HT: P3HT:PTB7 (PTB7—poly[[4,8-bis[(2-ethylhexyl)oxy]benzo[1,2-*b*:4,5-*b’*]dithiophene-2,6-diyl][3-fluoro-2-[(2-ethylhexyl)carbonyl]thieno[3,4-*b*]thiophenediyl]]) and P3HT:TQ1 (TQ1—poly[2,3-bis-(3-octyloxyphenyl)quinoxaline-5,8-diyl-*alt*-thiophene-2,5-diyl]) (Figure 11). F_4_TCNQ was used as a *p*-type dopant for both blends. The study’s main goal was to find the optimal mass ratio of the ingredients of the blends [32]. Implying a constant mass ratio, the Seebeck coefficient of both blends exceeded 1000 μV∙K^−1^. For the P3HT:PTB7 ratio of 10:90, the thermoelectric power of the device based on this blend was approximately 1100 μV∙K^−1^, but the P3HT:TQ1 blend could achieve a higher Seebeck coefficient value. For a P3HT:TQ1 ratio of 5:95, the thermoelectric power was 2000 μV∙K^−1^. The *HOMO* levels of P3HT and TQ1 were −5.00 and −5.60 eV, respectively, while PTB7 was approximately –5.15 eV. The *LUMO* level of the dopant was 5.24 eV. The better performance of the P3HT:TQ1 blend was due to the bandgap between the dopant and polymers [32].

Metal-core organic compounds may act as dopants; Hynynen’s group’s and Untilova’s group’s studies serve as examples. In both cases, P3HT (Figure 11) had the main role in the system, when molybdenum tris(dithiolene) complex (Mo(tfd-COCF_3_)_3_) (Figure 12) was the *p*-type dopant. Hynynen et al. focused on the mechanical and thermoelectric properties of P3HT films made by the tensile drawing technique. The group observed a good storage modulus of approximately 0.4 GPa. The Seebeck coefficient of this system was 112 μV∙K^−1^, while the electrical conductivity was 12.7 S∙cm^−1^. As a result, the power factor was equal to 16 μW∙m^−1^∙K^−2^ [73]. Nevertheless, Untilova et al. decided to improve the thermoelectric parameters of those films by high-temperature rubbing. This technique was used to obtain optimally aligned films to improve film conductivity. In so doing, the electrical conductivity of Mo(tfd-COCF_3_)_3_ doped P3HT after the high-temperature rubbing was equal to 509 S∙cm^−1^. The Seebeck coefficient decreased to 56 μV∙K^−1^, and the power factor increased 10 times and achieved 160 μW∙m^−1^∙K^−2^. In this case, the high-temperature rubbing enabled a change in the orientation of the molecules, which improved the thermoelectrical performance through a significant increase in electrical conductivity [74].

Polymer compounds can self-dope by groups containing active atoms, such as potassium or sodium. For example, conjugated polyelectrolytes (CPEs) (Figure 13) can be used as self-doping thermoelectric materials. A study analyzed a CPE with thiophene units connected with a ring and benzothiazole units in the main chain. The self-doping goal was to achieve higher conductivity, and thus, better performance reflected by PF because the ion used was a charge carrier [75]. However, CPE-TBA (TBA—*tert*-butyl ammonium) was characterized by high electric resistance and electrical conductivity of less than 10^−4^ S∙cm^−1^, which the researchers did not analyze further.

Nevertheless, the rest of the group—CPE-Na, CPE-K, CPE-C3-Na, and CPE-C3-K—showed strong electrical performance, with conductivities 0.16, 0.024, 0.22, and 0.048 S∙cm^−1^, respectively. Sodium derivatives had higher conductivities due to the smaller atom radius, allowing higher charge mobility. The Seebeck coefficient values of all four compounds were almost similar. Every polyelectrolyte could contribute to the Seebeck effect with a Seebeck coefficient value of approximately 200 μV∙K^−1^. In such a condition, the PF was determined by conductivity, and the highest PF of 0.84 μW∙m^−1^∙K^−2^ was achieved for CPE-C3-Na. The thermal conductivities of the studied compounds were approximately 0.2–0.3 W∙m^−1^∙K^−1^. It appears that the ion did not change the thermal conductivity of the polymer. ZT was not calculated due to uncertainty about combining in-plane electric parameters with out-of-plane thermal conductivity [75]. The parameters of CPEs observed by Mai et al. were not promising enough, so the authors searched for solutions to enhance the properties of polyelectrolytes. One of the solutions used by the Mai group was the synthesis of copolymers, i.e., CPE-K and CPE-TEG (Figure 13), where TEG was tetraethylene glycol units. The authors noted that the morphology of the thin film could be improved. For measurement, Mai et al. prepared four copolymers with different CPE-K and CPE-TEG ratios and compared them to pure CPE-K [76]. The prepared copolymers achieved a similar Seebeck coefficient value of approximately 230 μV∙K^−1^. The thermoelectric performance of these copolymers was influenced by electrical conductivity and differed for each compound. As mentioned earlier, the conductivity of pure CPE-K was 0.024 S∙cm^−1^, with a PF of 0.13 μW∙m^−1^ K^−2^. The synthesized copolymers showed higher conductivities than pure CPE-K, with the conductivity of CPE-K80 found to be the highest (0.44 S∙cm^−1^). The PF of CPE-K80 was 2.33 μW∙m^−1^∙K^−2^. The parameters of two other copolymers were also better than CPE-Na from previous research: the PFs of CPE-K90 and CPE-K70 were over 1 μW∙m^−1^∙K^−2^. The TEG side chains influenced the crystallization and charge mobility of the copolymers, and thus, were responsible for enhancing their parameters [76].

The poly(K_x_[Ni-ett])-based device was not the only one to show the Peltier effect. The Nafion^®^ (Figure 14) membrane cell with hydrogen electrodes was shown to achieve thermoelectric cooling. This cell is a polytetrafluorethylene (also known as Teflon^®^) derivative with a sulphone group as a side chain [77].

The Nafion cell ((C,Pt)H_2_(p^a^,T^a^)|H_2_O(p_w_^a^,T^a^)|m|H_2_(p^c^,T^c^)| H_2_O(p_w_^c^,T^c^)|(Pt,C)) achieved a Seebeck coefficient value of 670 μV∙K^−1^ at 340 K and 1 bar of pressure. In addition, the cell also exhibited thermoelectric cooling. However, the primary disadvantage with this cell was difficulty in water physical state control, and as a result, only the heat generated by the cathode was measurable. The emitted heat of the cathode was approximately 6 kJ mol^−1^ when voltage was applied to the cell [77].

As mentioned above, molecular junctions were mainly prepared using small molecules (e.g., benzene-1,4-diamine). However, they could also be prepared with polymeric materials, such as polyphenyl ether (PPE). Au–PPE–Au is an example of a molecular junction (Figure 15) [78,79]. The Seebeck coefficient relies on the number of repeating PPE units because the longer the main chain, the more suitable the polymer for thermoelectric applications. For *n* = 7, the Seebeck coefficient was calculated to be approximately 1000 μV∙K^−1^, and the dimensionless ZT was over 4. The ZT of the Au–PPE–Au junction was found to be close to 50 for *n* = 30, which suggests that this has promising applications in thermoelectrics in the future [78,79].

## 4. Conclusions

Organic thermoelectric materials and devices are the future of organic electronics. Small energy-demanding devices and sensors are crucial to the Internet of Things strategy. This review has presented various materials and their thermoelectric properties, which could possibly be regulated by structural modifications. Parameters such as the Seebeck coefficient and electrical and thermal conductivities play key roles in evaluating novel organic materials. The primary component of a thermoelectric device is a dopant, which determines the material’s conductivity. For example, P3HT-b-P(NDI2OD-T2) confirms carrier regulation, depending on the type of dopant used. Not only dopants but also exciplexes can act as efficient electricity carriers in flexible devices. These carriers could be successfully used in novel OLED devices to control temperature or harvest energy and reuse it. GSE is another factor that plays an essential role in developing thermoelectric devices, mainly based on carbon materials such as CNTs and fullerene derivatives. However, some studies have shown that other organic compounds (e.g., C_10_DNTT and BP) were also able to maintain similar parameters of CNT and fullerene devices or had slightly higher values of parameters. Nonetheless, there is a need to search for new materials. CNTs relying on GSE were successfully tested as materials for application as wearable electric generators. The CNT-based devices showed efficient performance, so if BP or C10DNTT replaced CNTs, the properties of wearable thermoelectric harvesters could be enhanced. Moreover, a highly conductive material such as PEDOT:PSS may be used in flexible devices, which can be used as wearable generators and produce a high amount of power due to its high figure of merit, which exceeds 0.3, depending on preparation methods. Every material has its advantages and disadvantages, but it cannot be denied that the development of OTEs is faster than it was earlier. As the IDTechEx report states, there is plenty of room for improvement in parameters to achieve the practical application of elastic thermoelectrics within the next ten years.

## Figures and Tables

**Figure 1 molecules-27-01016-f001:**
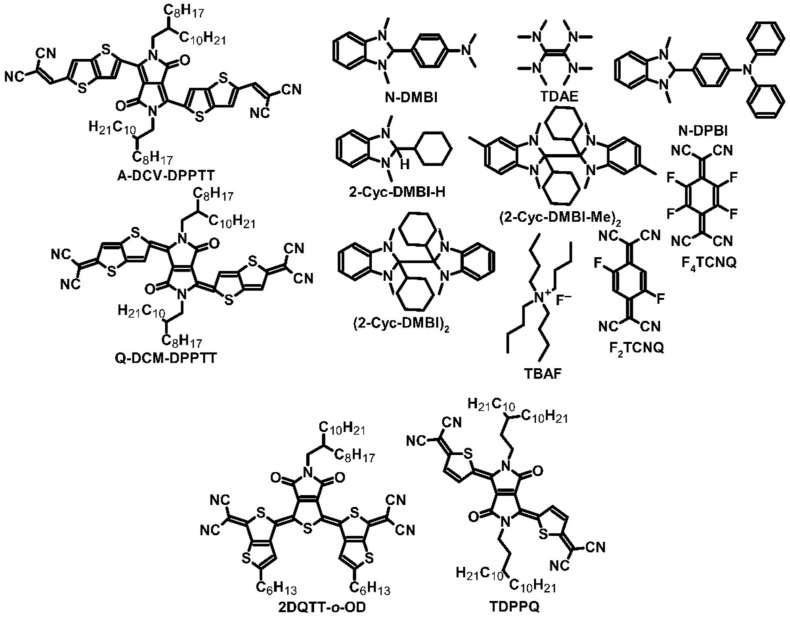
Structures of small molecules with differently distributed double-bond systems with respective dopants.

**Figure 2 molecules-27-01016-f002:**
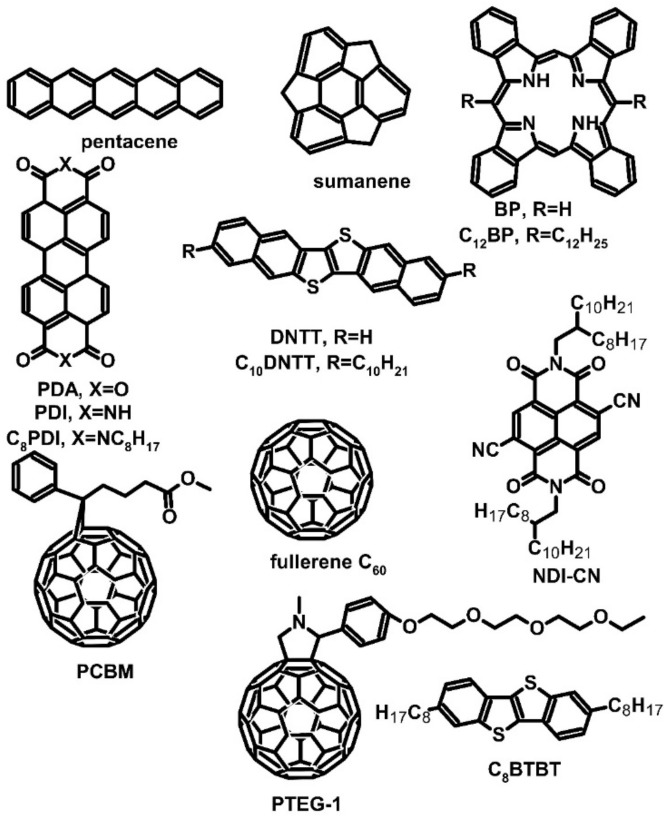
Structures used as organic materials or as giant Seebeck effect materials.

**Figure 3 molecules-27-01016-f003:**
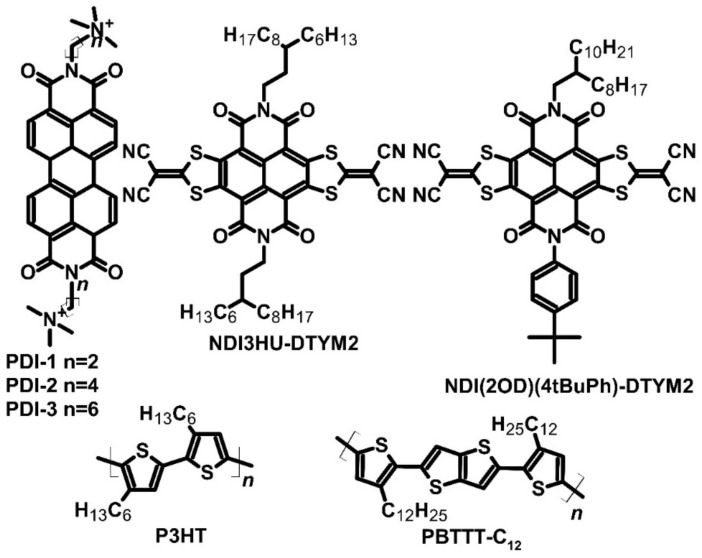
Structures of diimides and their respective polymers.

**Figure 4 molecules-27-01016-f004:**
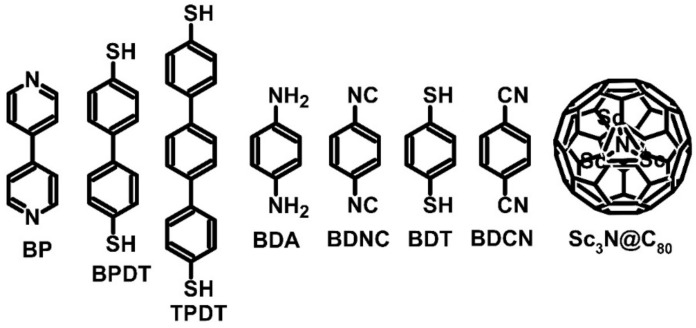
Examples of small molecule structures used as molecular junctions.

**Figure 5 molecules-27-01016-f005:**
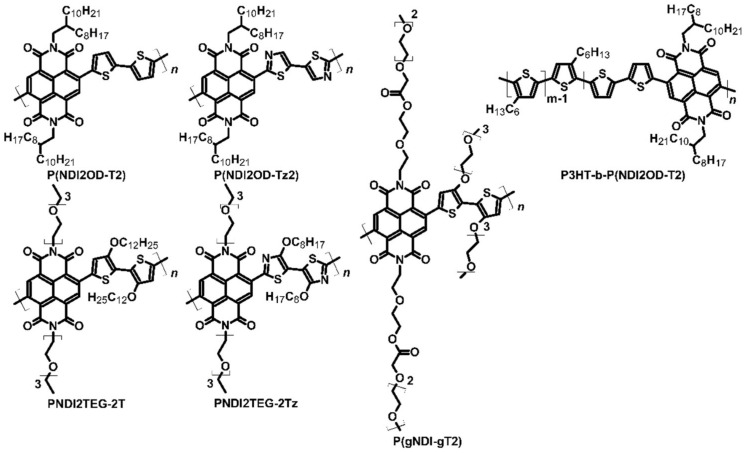
Examples of polydiimide structures used as active layers in OTEs.

**Figure 6 molecules-27-01016-f006:**
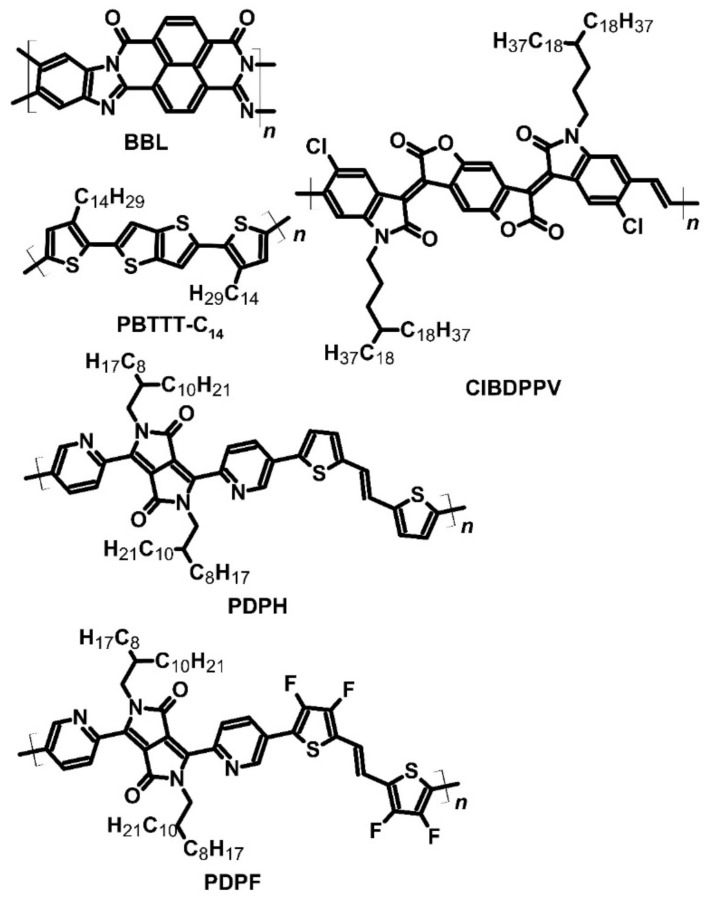
Structures of conducting polymers used as thermoelectric materials.

**Figure 7 molecules-27-01016-f007:**
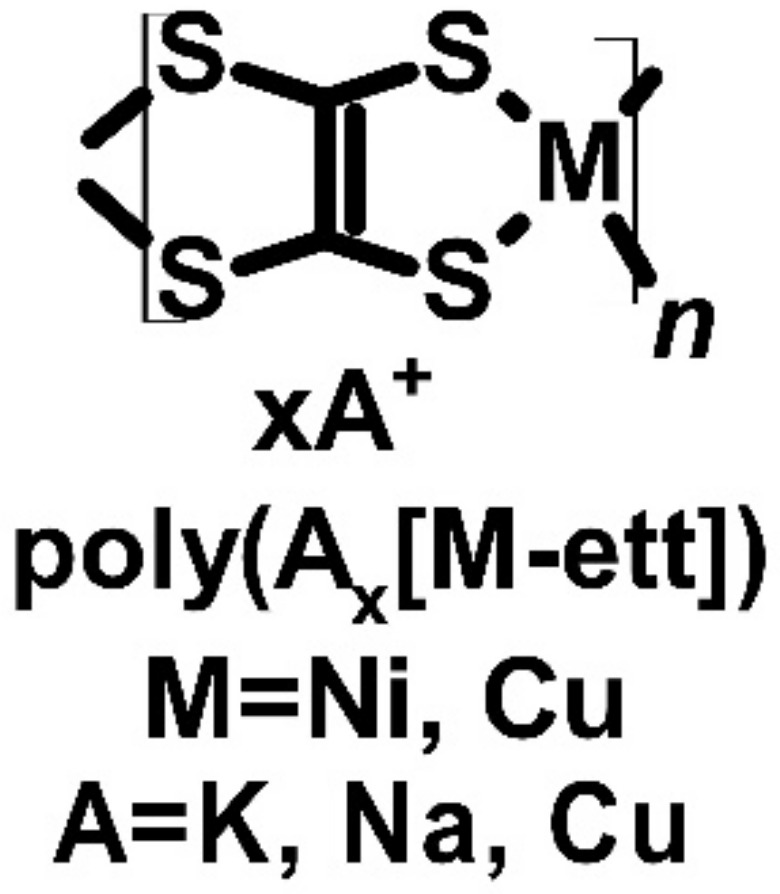
Structures of metallic core polymers.

**Figure 8 molecules-27-01016-f008:**
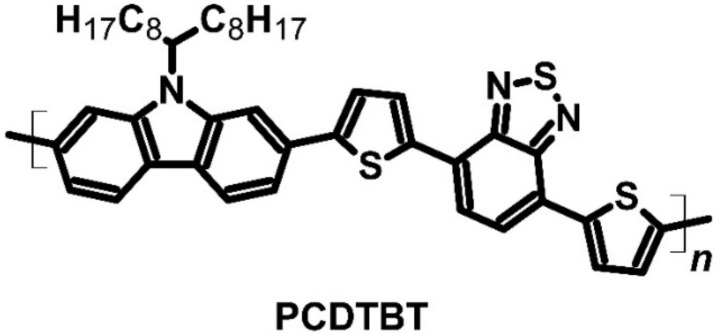
Structure of the carbazole-based polymer.

**Figure 9 molecules-27-01016-f009:**
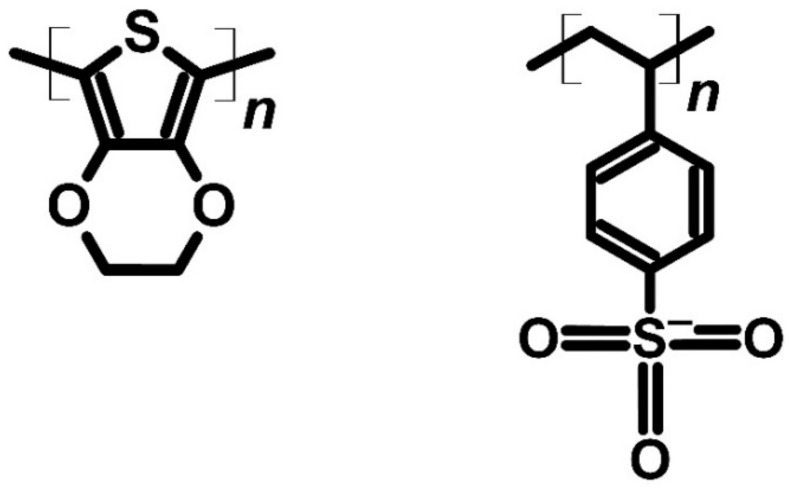
Structure of PEDOT:PSS.

**Figure 10 molecules-27-01016-f010:**
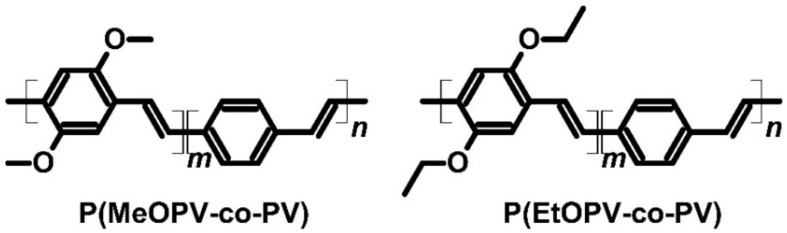
Structures of PPV-based conducting copolymers.

**Figure 11 molecules-27-01016-f011:**
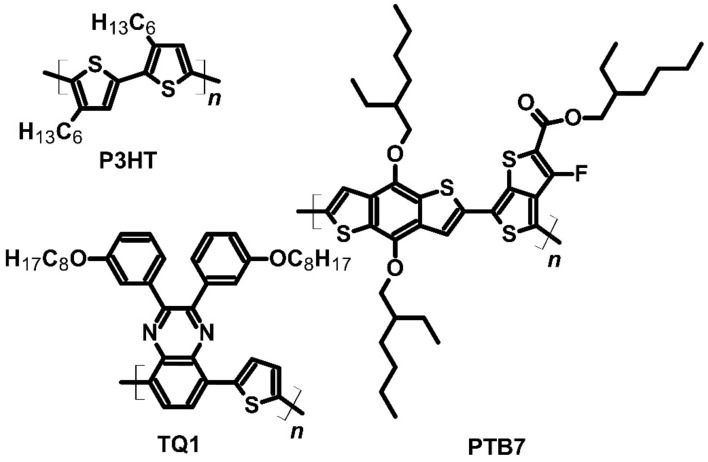
Structures of the blend ingredients.

**Figure 12 molecules-27-01016-f012:**
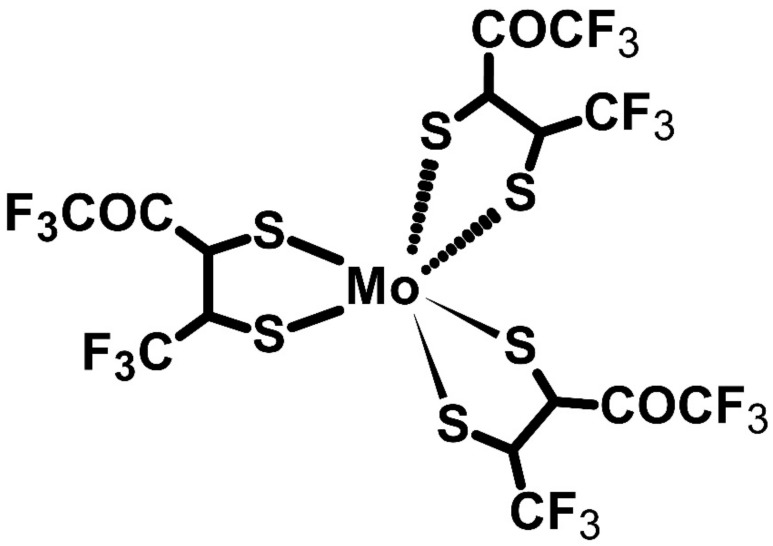
Structure of the Mo(tfd-COCF_3_)_3_.

**Figure 13 molecules-27-01016-f013:**
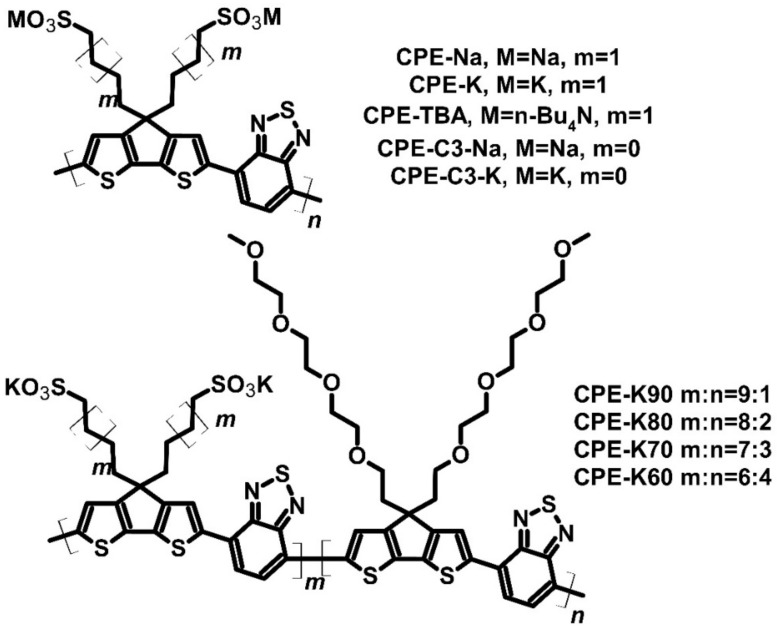
Structures of the conjugated polyelectrolytes.

**Figure 14 molecules-27-01016-f014:**
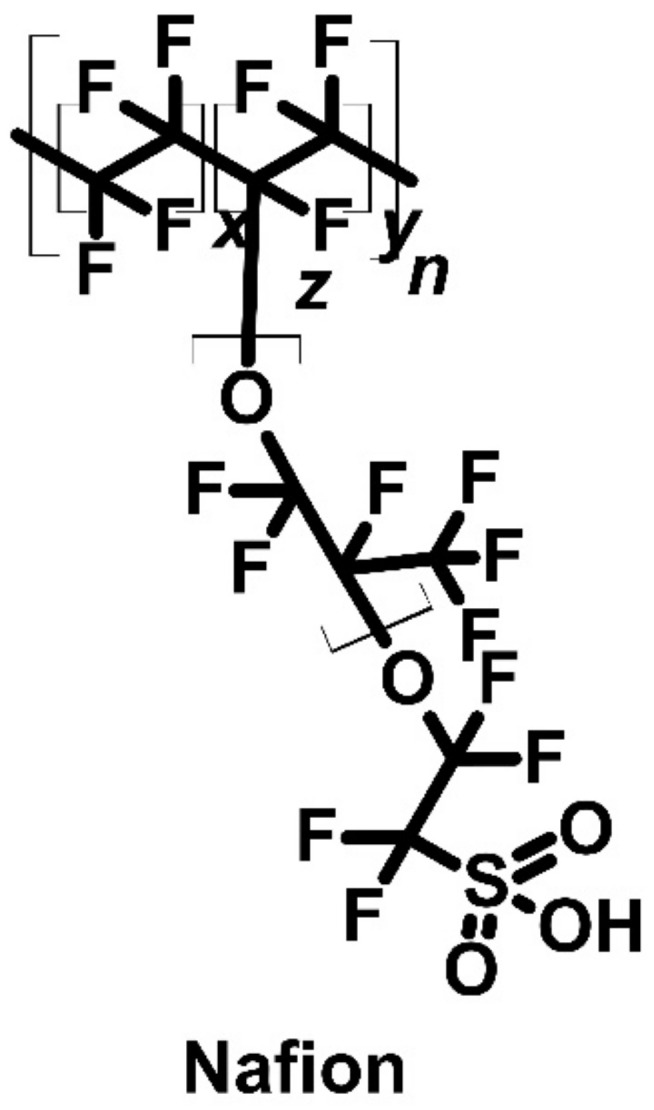
Structure of Nafion^®^ used as membrane cell.

**Figure 15 molecules-27-01016-f015:**
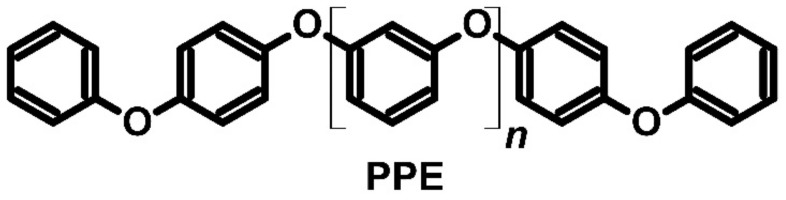
Structure of polyphenyl ether (PPE).

**Table 2 molecules-27-01016-t002:** Thermoelectric parameters of *p*-type small-molecule materials.

Material	Dopant	Cond. Type	*HOMO* [eV]	*LUMO* [eV]	*σ* [S/cm]	*α* [μV/K]	*σα^2^* [μW/(mK^2^)]	ZT	Ref.
Pentacene	I_2_	P	−4.61	−2.40	~50	40–70	20		[39]
Sumanene						~3 × 10^4^			[29]
BP			−4.70	−2.28		~1 × 10^5 (a)^	~1.5	5 × 10^−3^	[29]
C12BP						~8 × 10^4^			[29]
Pentacene		P	−4.61	−2.40		~4.5 × 10^4^			[29]

^(a)^ 360 K.

**Table 3 molecules-27-01016-t003:** Thermoelectric parameters of *n*-type polymer materials.

Material	Dopant	Cond. Type	*HOMO* [eV]	*LUMO* [eV]	*σ* [S/cm]	*α* [μV/K]	*σα^2^* [μW/(mK^2^)]	ZT	Ref.
P(NDI2OD-T2)	TDAE	N	−5.46	−3.99	3 × 10^−3^	−208	0.013		[46]
P(NDI2OD-Tz2)	TDAE	N	−5.90	−4.10	0.06	−447	1.5		[47]
PNDI2TEG-2Tz	N-DMBI	N	−5.56	−4.26	1.8	−159	4.6		[48]
P(gNDI-gT2)	N-DMBI	N	−4.83	−4.12	0.1	−200	0.4		[49]
P3HT-b-P(NDI2OD-T2)	N-DMBI	N	−5.20 −5.46	−3.99	1.7 × 10^−4^	−602	6.6 × 10^−3^		[50]
P(NDI2OD-T2)	N-DMBI	N	−5.46	−3.99	8 × 10^−3^	−850	6 × 10^−7^		[51]
P(NDI2OD-T2)	N-DPBI	N	−5.46	−3.99	4 × 10^−3^	−770	2 × 10^−7^		[51]
BBL	TDAE	N	−5.90	−4.00	1	−60	0.43		[46]
ClBDPPV	TBAF	N	−5.90	−4.30	0.62	−99.2	0.63	5.5 × 10^−4^	[52]
PDPF	N-DMBI	N	−5.82	−4.11	1.3	−235	4.65		[53]
poli(K_x_[Ni-ett])		N			44	−121.6	66	0.2 ^(a)^	[54,55,56]
poli(Na_x_[Ni-ett])		N			40	−75	22.5	0.042 ^(a)^ 0.1 ^(b)^	[54,55,56]

^(a)^ 300 K; ^(b)^ 440 K.

**Table 4 molecules-27-01016-t004:** Thermoelectric parameters of *p*-type polymer materials.

Material	Dopant	Cond. Type	*HOMO* [eV]	*LUMO* [eV]	*σ* [S/cm]	*α* [μV/K]	*σα^2^* [μW/(mK^2^)]	ZT	Ref.
PNDI2TEG-2T	N-DMBI	P	−5.39	−4.18	7 × 10^−4^	57.2	2.3 × 10^−4^		[48]
P3HT-b-P(NDI2OD-T2)	F_4_TCNQ	P	−5.20 −5.46	−3.99	1.4 × 10^−3^	196	5.5 × 10^−3^		[50]
PCDTBT	FeCl_3_	P	−5.50	−3.60	160	34	19		[62,63]
PEDOT:PSS		P	−5.20	−2.40	9.2 ^(a)^	15.7 ^(a)^	0.23 ^(a)^	4 × 10^−4 (a)^	[64]
PEDOT:PSS	5% DMSO	P	−5.20	−2.40	38.7 ^(b)^	12.0 ^(b)^	0.56 ^(b)^	1.9 × 10^−3 (b)^	[64]
PEDOT:PSS	10% DMSO	P	−5.20	−2.40	27.5 ^(c)^	13.5 ^(c)^	0.50 ^(c)^	1.35 × 10^−3 (c)^	[64]
PEDOT:PSS	5% EG	P	−5.20	−2.40	45.0 ^(c)^	11.8 ^(c)^	0.63 ^(c)^	1.65 × 10^−3 (c)^	[64]
PEDOT:PSS	10% EG	P	−5.20	−2.40	34.5 ^(b)^	12.3 ^(b)^	0.52 ^(b)^	1.47 × 10^−3 (b)^	[64]
PEDOT:PSS baked	5% DMSO	P	−5.20	−2.40	32.5 ^(b)^	12.3 ^(b)^	0.49 ^(b)^	1.4 × 10^−3 (b)^	[64]
PEDOT:PSS baked	10% DMSO	P	−5.20	−2.40	26.0 ^(a)^	11.6 ^(a)^	0.35 ^(a)^	8.1 × 10^−4 (a)^	[64]
PEDOT:PSS baked	5% EG	P	−5.20	−2.40	51.0 ^(d)^	11.6 ^(d)^	0.69 ^(d)^	1.75 × 10^−3 (d)^	[64]
PEDOT:PSS baked	10% EG	P	−5.20	−2.40	37.0 ^(a)^	12.0 ^(a)^	0.53 ^(a)^	1.25 × 10^−3 (a)^	[64]
PEDOT:PSS (pristine)	-	P	−5.20	−2.40	1–3	13–15	0.2		[65]
PEDOT:PSS (annealed in 220 °C)	-	P	−5.20	−2.40	596	23.3	32.5		[65]
PEDOT:PSS^f)^	DMSO	P	−5.20	−2.40	880	72	469	0.42	[66]
PEDOT:PSS^f)^	EG	P	−5.20	−2.40	890	62	345	0.28	[66]
PEDOT:PSS	CH_3_NO, H_2_SO_4_, NaBH_4_	P	−5.20	−2.40	1786	28.1	141		[67]
PEDOT:PSS	EG, PVA, Te-NWs	P	−5.20	−2.40	382.4	11.3	8.5		[68]
PEDOT	Bi_2_Te_3_	P	−5.20	−2.40	483	168	1350	0.58	[69]
PDPH	N-DMBI	P	−5.61	−3.93	1.01 × 10^−3^	71	5.11 × 10^−4^		[53]
PBTTT-C_14_	F_4_TCNQ	P	−5.10	−3.10	3.51 ^(e)^ 220 ^(f)^	60 ^(e)^ 39 ^(f)^	1.3 ^(e)^ 32 ^(f)^		[70]
PBTTT-C_14_	F_2_TCNQ	P	−5.10	−3.10	2 × 10^−3 (e)^36 ^(f)^	755 ^(e)^ 140 ^(f)^	0.11 ^(e)^ 70 ^(f)^		[70]
P(MeOPV-co-PV)	I_2_	P			183	43.5	34.6	0.014	[71,72]
P(EtOPV-co-PV)	I_2_	P			350	47	77.3	0.1	[71,72]
P3HTPTB7	F_4_TCNQ	P	−5.00 −5.15		~4 ^(f)^	1100 ^(h)^ ~130 ^(g)^	~7 ^(g)^		[32]
P3HTTQ1	F_4_TCNQ	P	−5.00 −5.60		~4 ^(f)^	2000 ^(i)^ ~130 ^(g)^	~7 ^(g)^		[32]
P3HT	Mo(tfd-COCF_3_)_3_	P	−5.00		12.7	112	16		[73]
P3HT	Mo(tfd-COCF_3_)_3_	P	−5.00		509	56	160		[74]
CPE-Na		P			0.16	165	0.44		[75]
CPE-K		P			0.024	230	0.13		[75]
CPE-TBA		P			<1 × 10^−4^				[75]
CPE-C3-Na		P			0.22	195	0.84		[75]
CPE-C3-K		P			0.048	200	0.19		[75]
CPE-K90		P			0.25	~230	~1.16		[76]
CPE-K80		P			0.44	~230	2.33		[76]
CPE-K70		P			0.30	~230	~1.66		[76]
poli(Cu_x_[Cu-ett])		P			9.5	83	6.5	2 × 10^−3 (j)^ 0.014 ^(k)^	[54,55,56]
Nafion membrane		P				670			[77]
Au-BPDT-Au		P				13.0			[35]
Au-TPDT-Au		P				15.7			[35]
Au-BDA-Au		P				2.2			[36]
Au-BDT-Au		P				2.4			[36]
Au-PPE-Au		P				1000		>4	[78,79]

^(a)^ 272 K; ^(b)^ 205 K; ^(c)^ 240 K; ^(d)^ 258 K; ^(e)^ spin coated; ^(f)^ vapor sublimed; ^(g)^ pure P3HT; ^(h)^ P3HT:PTB7—10:90 of mass ratio; ^(i)^ P3HT:TQ1—5:95 of mass ratio; ^(j)^ 230 K; ^(k)^ 380 K.

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
