# Peer review of "Organic Thermoelectric Materials as the Waste Heat Remedy"

_molecules, 2022, doi:10.3390/molecules27031016_

Round 1

Reviewer 1 Report

This is a review about recent advances about organic thermoelectric materials. Some developments for both the organic small molecules and organic polymers have been included for thermoelectric applications. I think this is a good review essentially. However, the following should be revised.

1, The whole manuscript include the advances of thermoelectric materials, but NOT thermoelectric devices. Therefore, I suggest to delete the Keywords of "devices" and "generators". 

2, Page 3, in the title of section 2, I suggest to delete "Devices".

3, There are 3 sections totally, including Introduction, Organic Thermoelectric Materials, and Conclusions. I recommend to divide the sceond section into two parts of Orgnanic Small Molecular Thermoelectric Materials and Organic Polymer Materials. Furthermore, give some sub-divisions of each section.

4, Some recent publications of Organic Thermoelectric Materials are recommended: Nano Energy, 2021, 80, 105448; 

Compos. Commun., 2021, 27, 100855; 

Nat. Commun., 2018, 9, 3817; 

 Energy Mater. Adv., 2021, 2021, 572537.

Based on the above comments, I recommend revisions.

Author Response

Response to the Reviewer 1

Reviewer Comment: The whole manuscript include the advances of thermoelectric materials, but NOT thermoelectric devices. Therefore, I suggest to delete the Keywords of "devices" and "generators".

Response: We agree with the Reviewer’s comment. As a result, we decided to change keywords to more appropriate ones.

Reviewer Comment: Page 3, in the title of section 2, I suggest to delete "Devices".

Response: We see that comment is connected with the previous one, so we decided to replace “Devices” by “Materials” in the title of the commented section.

Reviewer Comment: There are 3 sections totally, including Introduction, Organic Thermoelectric Materials, and Conclusions. I recommend to divide the second section into two parts of Organic Small Molecular Thermoelectric Materials and Organic Polymer Materials. Furthermore, give some sub-divisions of each section.

Response: We agree with Reviewer’s statement. We decided to divide the second section following your comment when n-type and p-type materials are subsections for each part.

Reviewer Comment: Some recent publications of Organic Thermoelectric Materials are recommended:

Nano Energy, 2021, 80, 105448;

Compos. Commun., 2021, 27, 100855; 

Nat. Commun., 2018, 9, 3817; 

Energy Mater. Adv., 2021, 2021, 572537.

Response: We would like to thank you for this suggestion, we decided to expand our review with proposed articles.

Reviewer 2 Report

Manuscript ID: molecules-1550928

Comments to the Authors:

In this manuscript, the authors discussed the developments made in thermoelectric materials, including small molecules, polymers, molecular junctions, and their application as TEGs and/or TECs. The research work represents scientific-interesting, but the research content lacks sufficient explanation. The manuscript can be improved if the authors can revise the following points:

  1. The authors should require highlighting the main objectives of the manuscript in the introduction section.
  2. Please raise the readability of this manuscript by a substantial explanation of the results and discussion.
  3. The authors concluded that CNTs as promising material for the developing thermoelectric devices. Please, it should explain clearly within justification.
  4. The list of references should be expanding.

Author Response

Response to the Reviewer 2

Reviewer Comment : The authors should require highlighting the main objectives of the manuscript in the introduction section.
Response: We decided to add sentences covering your comment.

Reviewer Comment : Please raise the readability of this manuscript by a substantial explanation of the results and discussion.

And

The authors concluded that CNTs as promising material for the developing thermoelectric devices. Please, it should explain clearly within justification.

Response: We would like to thank the Reviewer, we updated the manuscript to increase the readability.

Reviewer Comment : The list of references should be expanding.

Response: We agree with the Reviewer comment and we decided to add more recent publications to our manuscript.

This manuscript is a resubmission of an earlier submission. The following is a list of the peer review reports and author responses from that submission.

Round 1

Reviewer 1 Report

See annex

Author Response

We are grateful to the reviewer for their valuable comments and helpful suggestions. We have carefully considered all the recommendations and included our responses as described below. The sentences, including corrected parts, are highlighted in yellow in the revised manuscript.

Reviewer Comment : This manuscript collects data published on the thermoelectric properties of a vast number of organic semiconductors, encompassing small molecules and polymers doped with many different dopants. This is a good overview for specialists in the field. However, I strongly regret that this manuscript is purely descriptive and does not better describe the key electronic processes and does not provide at the end a discussion on the parameters favoring good thermoelectric properties.
Response: To be honest, our idea was to present more data information rather than description. In our opinion, there are too many scientific gaps in the case of organic thermoelectric. In many cases, the author’s interpretation is wrong or only limited to the particular material that doesn't work in the global picture, which means we still need more work in this area. We didn’t want to exclude that information but show different points of view. We want to give readers a source where they can get a particular set of information, but we are far from saying if that would improve the thermoelectric materials. There is enormous demand and perspective for organic thermoelectric but in our opinion also a lot of work to do in comparison to inorganic materials. Nevertheless, the same was for other organic electronic devices like OLEDs or OPVs.

Reviewer Comment : “I am disturbed by the title of the paper since thermoelectricity is not the only application guaranteeing the future of organic electronics; we can also cite organic photovoltaics, bioelectronics, organic white lighting among others. Throughout the reading of the paper, we can realize that the authors do not have a good fundamental knowledge of electronic processes in organic semiconductors, making altogether this paper of low scientific interest.

Response: We agree with the reviewer's statement about the title of this paper. Readers could read it as the only way for organic electronic future, which was not our intention. We decided to change this paper's title to avoid suggestions about the advantages of one type of organic electronics against another.

Reviewer Comment : “Line 106: However, due to the low concentration of radicals, which can be explained by missing ππ stacking and increased d-spacing distance, Q-DCM-DPPTT exhibited lower conductivity. What is the link between the concentration of radicals and the stacking ? Increased d-spacing should favor the insertion of the dopants !

Response: We cannot agree with the reviewer, I understand the concept, but this is the common problem with conjugated compounds, especially conjugated polymers. There are several conductivity paths in conjugated compounds, one of them is also a hooping mechanism. Of course, a higher distance would allow for more dopants, but higher doping doesn’t mean higher conductivity. At some point, the conductivity will be limited by the dopant, not the host. In this case, we have a nice example of the strong interaction (ππ stacking), which is typical for highly conjugated materials and increased hooping mechanism. Of course, there are pros and cons of the ππ stacking, but in this case, it shows excellent that ππ stacking improve charge mobility.

Reviewer Comment : “Line 116 : Thermoelectric efficiency is determined only by dopant concentration. This is a very strong statement since the film morphology is for instance as primordial as the nature of the dopant.”

Response: The following statement was an oversimplification. We corrected that part. Thermoelectric efficiency is determined not only by dopant concentration was the correct one.

Reviewer Comment : “Line 155 : Bismuth can be considered as an effective interfacial dopant, due to its ideal molecular pack-ing that allows loading increased concentration of carrier material [25]. It is rather odd to talk about the molecular spacing of bismuth !”

Response: I think we use an oversimplification of the sentence. We want to say that Bismuth is good as a dopant and fits weel molecular packing. Therefore, we changed the sentence: “Bismuth can be considered an effective interfacial dopant, due to Fermi level shift. Because of that, it was possible to observe the change of HOMO level from 2.58 (pristine TDPPQ) to 2.73 eV below Fermi level (TDPPQ with 3nm of bismuth layer). Moreover, there was appearing possible gap state around 0.60 eV below Fermi level which allows to the easier movement of charge between HOMO and LUMO levels.”

Reviewer Comment : “Line 184 : The highest thermal conductivity was observed for BP, with a value of over 10–6 184 S∙cm– 1 , and thus this material was characterized by the highest PF compared to other in- 185 vestigated materials. There is no apparent link between the thermal conductivity and PF; moreover, high thermal conductivity is highly detrimental for ZT.

Response: Following statement was our writing mistake. We meant about electrical conductivity (which directly links with PF) not the thermal one, as the states unit written in the mentioned fragment (S∙cm– 1).

Reviewer Comment : “Line 229 : NDI-CN acted as an agent that modifies the density of state, rather than as a dopant. All dopants modify the density of states !?”

Response: From a material point of view, yes, all dopants can affect the DOS, but all materials act as dopants, and this is the case. We modified the sentence NDI-CN due to the aggregation acted as an agent that modifies the density of state only not as a dopant, and negatively influenced conductivity.”

Reviewer 2 Report

  1. In the Introduction section, basic history of organic thermoelectrics was briefly mentioned, however, I think basics and research issues of organic thermoelectrics should be included in detail.
  2. Section 2.2. and caption of Figure 2 : I recommend using organic materials rather carbon materials. “Carbon materials” can also indicate carbon nanotubes, graphene and so on.
  3. In Table 2 and manuscript: it would be better if n-type and p-type researches are categorized.
  4. Section 3.3. : other conducting polymers should include sentences to mention PEDOT:PSS, even though you exclude review of researches of PEDOT:PSS.
  5. If possible, include more current researches in the field of organic thermoelectrics.
  6. English should be improved.

Author Response

Responses to Reviewer’s Comments

We are grateful to the reviewer for their valuable comments and helpful suggestions. We have carefully considered all the recommendations and included our responses as described below. The sentences, including corrected parts, are highlighted in yellow in the revised manuscript.

Reviewer Comment: “ In the Introduction section, basic history of organic thermoelectrics was briefly mentioned, however, I think basics and research issues of organic thermoelectrics should be included in detail.

Response: We agree with your comment. And we decided to write in detail issues connected with the origins of organic thermoelectric generators. We found interesting connections between the first thermoelectric devices and pioneers of conducting polymers.

Reviewer Comment: “Section 2.2. and caption of Figure 2 : I recommend using organic materials rather carbon materials. “Carbon materials” can also indicate carbon nanotubes, graphene and so on.

Response: Despite mentioning in that part of manuscript fullerenes, which may be considered carbon material, we agree with your statement and decided to change the mentioned fragment into “organic materials”. Due to changes in the manuscript's structure connected with the reviewer's next comment, we made changes only in the caption of Figure 2.

Reviewer Comment: “ In Table 2 and manuscript: it would be better if n-type and p-type researches are categorized.

Response: We agree with your point of view connected with the categorization of n-type and p-type researches and changed the manuscript to keep with that categorization which is better than the original one.

Reviewer Comment: “Section 3.3. : other conducting polymers should include sentences to mention PEDOT:PSS, even though you exclude review of researches of PEDOT:PSS.”

Response: PEDOT:PSS is one of the most important conducting polymer used in organic electronics, and we decided to put first and recent researches on this compound.

Reviewer Comment
: “If possible, include more current researches in the field of organic thermoelectrics.”

Response: We decided to connect this comment with the previous one about PEDOT:PSS, so it was possible to include more recent researches.

Reviewer Comment: “English should be improved.”

Response: We corrected the English in the manuscript, although we had only 7 days for the review.

Round 2

Reviewer 1 Report

The small changes and reply made by the authors did not change my opinion about the manuscript. Some wrong statements have been corrected but the answers made to other comments remain quite confusing.